# Nitrous oxide as a function of oxygen and archaeal gene abundance in the North Pacific

Mark Trimmer[1], Panagiota-Myrsini Chronopoulou[1], Susanna T. Maanoja[1], Robert C. Upstill-Goddard[2], Vassilis Kitidis[3] & Kevin J. Purdy[4]

Oceanic oxygen minimum zones are strong sources of the potent greenhouse gas $N_2O$ but its microbial source is unclear. We characterized an exponential response in $N_2O$ production to decreasing oxygen between 1 and 30 $\mu$mol $O_2 l^{-1}$ within and below the oxycline using $^{15}NO_2^-$, a relationship that held along a 550 km offshore transect in the North Pacific. Differences in the overall magnitude of $N_2O$ production were accounted for by archaeal functional gene abundance. A one-dimensional (1D) model, parameterized with our experimentally derived exponential terms, accurately reproduces $N_2O$ profiles in the top 350 m of water column and, together with a strong $^{45}N_2O$ signature indicated neither canonical nor nitrifier–denitrification production while statistical modelling supported production by archaea, possibly via hybrid $N_2O$ formation. Further, with just archaeal $N_2O$ production, we could balance high-resolution estimates of sea-to-air $N_2O$ exchange. Hence, a significant source of $N_2O$, previously described as leakage from bacterial ammonium oxidation, is better described by low-oxygen archaeal production at the oxygen minimum zone's margins.

[1] School of Biological and Chemical Sciences, Queen Mary University of London, London E1 4NS, UK. [2] School of Marine Science and Technology, Ridley Building, University of Newcastle, Newcastle upon, Tyne NE1 7RU, UK. [3] Plymouth Marine Laboratory, Prospect Place, West Hoe, Plymouth PL1 3DH, UK. [4] School of Life Sciences, University of Warwick, Coventry CV4 7AL, UK. Correspondence and requests for materials should be addressed to M.T. (email: m.trimmer@qmul.ac.uk).

Permanent oceanic oxygen minimum zones (OMZs) are significant sources of tropospheric $N_2O$ ($\sim 0.8–1.35$ Tg N yr$^{-1}$ or 20–75% of oceanic total[1] excluding coasts[2]), a potent greenhouse gas that also plays key roles in atmospheric chemistry[1,3]. Oversaturation of $N_2O$ within an OMZ is undoubtedly due to microbial activity but the precise nature of the organisms and biochemistry responsible for its production remain to be fully characterized. Some $N_2O$ production in OMZs has been ascribed to classic, canonical denitrification at the base of an oxycline[4,5], whereas deeper into the functionally anoxic core of an OMZ, there is also net reduction of $N_2O$ to $N_2$ by denitrification[5–7]. Above the anoxic core, production of $N_2O$ is traditionally described as a single function of bacterial nitrification under oxygen stress, with the yield of $N_2O$, from the oxidation of ammonium, increasing as oxygen declines[8–10]. More recently, there have also been suggestions for a coupling (both inter- and intracellular) between denitrification and nitrification as a means of $N_2O$ production and there is growing evidence for a direct contribution from the Archaea to this process[2,4,11–13].

The documented thickening of OMZs across the world[14] has not only increased the volume of low oxygen waters with a potential to produce $N_2O$ but such thickening also makes that $N_2O$ more readily exchangeable with the atmosphere. There is clearly therefore a need to improve our understanding of the production of this atmospherically potent $N_2O$ at both the margins of OMZs and beyond in hypoxic, coastal waters[2]. Many have taken the linear negative correlation often observed between $N_2O$ (oversaturation relative to atmospheric equilibration) and $O_2$ in surface waters to indicate bacterial nitrification as the predominant source of $N_2O$ (that is, the $N_2O$ anomaly versus apparent oxygen utilization and see ref. 15). Classic bacterial nitrification as the source of $N_2O$ was corroborated by early observations with a pure culture of the ammonia-oxidizing bacterium Nitrosomonas sp., where the yield of $N_2O$ per mole of ammonium oxidized increased exponentially as oxygen declined[8]. This regulatory effect of $O_2$ on $N_2O$ production in the ocean is now widely accepted (some 100 papers citing[8] in relation to ocean $N_2O$ production), though its exponential form has not, to the best of our knowledge, been characterized experimentally in the ocean below 30 μmol $O_2$ l$^{-1}$ (ref. 13). In addition, there are few, if any, ocean-based experimental data to substantiate this single physiological response. For example, incubation of OMZ oxycline waters with $^{15}NH_4^+$ might be expected to yield predominantly $^{46}N_2O$, that is, both N in $N_2O$ derived from $NH_4^+$, ($^{15}NH_4^+ \rightarrow {}^{15}NH_2OH \rightarrow {}^{15}NO + {}^{15}N_2O \rightarrow {}^{15}NO_2^-$), if classic oxygen-stressed, bacterial-nitrifier $N_2O$ production was active, but this is not the case[4]. Rather, this pathway of bacterial-nitrifier $N_2O$ production has routinely been used for the purposes of mass balance or to simply rationalize water column distributions of $N_2O$ without any supporting experimental evidence[9,10].

In addition to the poorly substantiated mechanistic basis for bacterial-nitrifier $N_2O$ production in the ocean, it is now evident that the Archaea are widespread in the ocean, playing significant roles in key processes such as nitrification and potentially the production of $N_2O$ (refs 12,13,16,17). The Thaumarchaeota, the archaeal phylum that encompasses the ammonium oxidizing archaea (AOA[18]), are commonly found in low-oxygen waters at the margins of an OMZ. AOA abundance decreases as oxygen concentrations rise towards air saturation in the upper mixed layers of the ocean[13,19] and have been shown to also decrease as oxygen practically disappears at the oxic to anoxic interface at the core of an OMZ[6]. Thus, lower-oxygen waters appear to be an important niche for at least some Thaumarchaeota groups[20]. It is well established, at least for laboratory cultures, that bacterial ammonia oxidizers can produce $N_2O$ directly: either as a by-product of nitrification[8] or through nitrifier–denitrification[21].

It is now evident that some AOA can also produce $N_2O$ during nitrification, but probably not via a nitrifier–denitrification-like process. New models of archaeal ammonia oxidation indicate a key role for NO produced by nitrite reductase encoded by the gene AnirK[22] providing a clear link to the production of $N_2O$, possibly via hybrid $N_2O$ formation[22–25]. A role for AOA-mediated $N_2O$ production has been suggested in the oceans, with process, natural abundance isotope values and molecular data supporting this idea[12,20,26,27]. There has, however, been no formal experimental characterization of $N_2O$ production at oxygen concentrations representative of the margins of an OMZ ($< 62.5$ μmol $O_2$ l$^{-1}$) where $N_2O$ accumulates[2,28] and/or the abundance of AOA (or any other candidate organisms) in representative samples of the ocean[13].

Here we provide experimental evidence, with samples from the eastern tropical North Pacific (ETNP), which clearly links an exponential increase in $N_2O$ production to decreasing oxygen between 1 to 30 μmol l$^{-1}$ and to archaeal gene abundance, together with a $^{15}N$ pattern in the $N_2O$ that is reconciled by an archaeal mode of production.

## Results

**Water column characteristics**. Along our offshore transect through the nitrite maximum zone (NMZ, Fig. 1a and Supplementary Table 1), the mixed layer depth (MLD) extended down to approximately 20 to 25 m and then the density increased steadily to a sharp inflection at 35 to 40 m, marking the base of the pycnocline (Fig. 1b). Oxygen dropped rapidly in the pycnocline to $\leq 50$ μmol $O_2$ l$^{-1}$ at its base but remained above the limit of detection for the Seabird Sensor to approximately 340 m (Fig. 1c). Within these low-oxygen waters (6.8 μmol $O_2$ l$^{-1}$, median value) we measured a broad, characteristic peak in $N_2O$ of up to 104 nmol $N_2O$ l$^{-1}$ (Fig. 1d). Deeper, at around 350 m, oxygen became comparatively constant, with the functionally anoxic core of the OMZ[7], where both the secondary nitrite maxima and $N_2O$ minima were measured, occurring deeper still at 400 to 450 m (Supplementary Fig. 1a). We found that the waters at all of the depths described so far were supersaturated with $CO_2$ (Supplementary Fig. 1b), with $CO_2$ being strongly correlated with $N_2O$ over the top 45 m (Supplementary Fig. 1c).

We set our observations for $N_2O$ into the wider context of the eastern tropical North Pacific by comparing them with profiles in the MEMENTO database[29] (Supplementary Fig. 2). Although there is considerable variation in the profiles, peak concentrations of 60 to 100 nmol $N_2O$ l$^{-1}$ at 100 m are present between approximately 0 °N to 22 °N and out to approximately 155 °W, with this triangle roughly marking the extent of the nitrite maximum zone (NMZ), within the wider boundary of the OMZ as a whole[30].

**Nitrous oxide production as a function of oxygen**. We measured the production of $^{15}N_2O$ in incubations with $^{15}NO_2^-$ at two depths at each of our six offshore sites ($n = 12$ groups of experiments). To generate natural variation in ambient water column oxygen and nitrous oxide concentrations, each depth was either within or beneath the oxycline (Supplementary Table 2). Each group of experiments comprised up to six oxygen treatments, giving us 70 independent observations ($n = 70$) for the production of $N_2O$ as a function of oxygen (Table 1). Production of $N_2O$ ($pN_2O_{total}$ equations 1–5) was strongly modulated by the level of oxygen in each treatment (likelihood ratio test for treatment, degree of freedom 5, $\chi^2$ 38.365, $P < 0.0001$ (ref. 7) and degree of freedom 3, $\chi^2$ 34.688, $P < 0.0001$ for the full and pooled data sets, respectively) and was maximal in waters degassed with

nitrogen (Fig. 2a,b). In addition, the $^{15}N_2O$ produced in each experiment was predominantly single labelled $^{45}N_2O$ (that is, only having one $^{15}N$), at a level far above (81%, on average) that expected for denitrification (4–23%; Supplementary Table 1) given the $^{15}N$ labelling of the $NO_2^-$ pool (Fig. 2c, equations 2 and 3 in the 'Methods' section and Supplementary Table 1). Given the

labelling of produced $N_2O$, we could only ascribe 19% of the $N_2O$ to the reduction of exogenous nitrite ($N_2O_{exogenous}$, equation 1) with the large majority (81%) of the $N_2O$ being due to some form of endogenous coupling ($N_2O_{endogenous}$, equation 4 and see below).

Ambient controls were used to represent $N_2O$ production in unadulterated water, that is, straight from the sampling bottles on the conductivity-temperature-depth rosette and, in these, oxygen concentrations ranged naturally from 1 to 199 $\mu$mol $O_2$ l$^{-1}$. In addition, the concentration of oxygen set at each level of the treatment varied (coefficient of variation (CV) of 4% to 35% for all the levels) across the 12 groups of experiments (Table 1). To account for this oxygen gradient in the 12 groups of experiments, and any natural variation in the water samples, we used a nonlinear mixed-effects approach to model the production of total $N_2O$ as an exponential function of decreasing oxygen (Table 2 and Supplementary Fig. 3). The most parsimonious model (M2) required only a random intercept ($a$), which allowed the overall magnitude of total $N_2O$ production to vary randomly between the 12 groups of experiments, while keeping the response to oxygen ($b$) constant. The mixed-effects model captures the data well (Fig. 3). The overall exponential increase in production of $N_2O$ with oxygen decreasing below 30 $\mu$mol $O_2$ l$^{-1}$ is not only consistent with $N_2O$ accumulating below 30 $\mu$mol $O_2$ l$^{-1}$ in the water column (Fig. 1d) but also with distributions seen in many parts of the tropical North Pacific (as above, Supplementary Fig. 2). We also measured the production of $N_2O$ over time at two oxygen concentrations (Supplementary Fig. 4) to check whether our 72 h incubation overestimated production. Where production was strongest (30 and 56 nmol m$^{-3}$ d$^{-1}$) and representative of the 12 main experiments (median 58 nmol m$^{-3}$ d$^{-1}$, <30 $\mu$mol $O_2$ l$^{-1}$), it was approximately linear over the first 18 h and then decreased over time. If anything, our single time point incubations may have underestimated $N_2O$ production slightly. Overall, however, we conclude that our experiment captured the regulation of $N_2O$ production by oxygen in the ocean. None of the incubations produced any $^{15}N$ labelled $N_2$, not even at 1 $\mu$mol $O_2$ l$^{-1}$.

Nitrification was also clearly active in the water column. We measured a primary nitrite maximum (Supplementary Fig. 1a) and, in ambient samples of water, with oxygen at 1 to 23 $\mu$mol $O_2$ l$^{-1}$, significant oxidation of both ammonium and nitrite (2.4 nmol N l$^{-1}$ d$^{-1}$ and 19.1 nmol l$^{-1}$ d$^{-1}$ on average for each, respectively), and net nitrification of up to 8.2 nmol N l$^{-1}$ d$^{-1}$ (Supplementary Table 2).

**Variation in nitrous oxide production with gene abundance**. By allowing the magnitude of total $N_2O$ production ($a$) to vary randomly between the 12 groups of experiments the mixed-effects model was able to derive an overall 'population' estimate for the

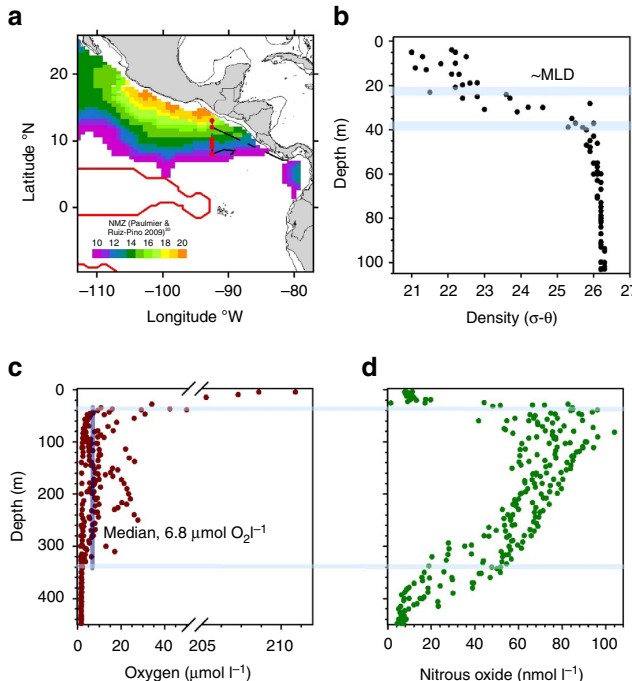

**Figure 1 | Open ocean OMZ sites and main water column profiles.**
(**a**) Offshore transect (bold black line with sites as filled red circles) through the nitrite maximum zone (NMZ) from 13 °N, at approximately 120 km off the coast, to 8 °N, at 670 km offshore. The red line marks the boundary of the permanent OMZ (min. $O_2 < 20$ $\mu$mol l$^{-1}$,) and the colour indicates the intensity of the nitrate deficit (>10 $\mu$mol l$^{-1}$) within the NMZ (reprinted from ref. 80). (**b**) Density as a function of depth: upper light-blue line, approximate mixed layer depth (MLD); and lower light-blue line, base of the pycnocline. (**c**) Oxygen dropped rapidly in the pycnocline to ≤50 $\mu$mol $O_2$ l$^{-1}$ at its base (upper light-blue line) but remained above the limit of detection for the Seabird Sensor (∼ 1.6 $\mu$mol l$^{-1}$) to 340 m (lower light-blue line). Within these boundaries (light-blue lines in **c,d**), oxygen was present at 6.8 $\mu$mol l$^{-1}$, on average (median, vertical dark-blue), and we measured a broad peak in $N_2O$ (**d**). Profiles to 700 m are given in Supplementary Fig. 1. Panels **b–d** were drawn in SigmaPlot (Systat Software, San Jose, CA, USA).

**Table 1 | Measuring the production of N₂O as a function of oxygen.**

| Treatment | p.p.m. | Balance | Final in water (µM) | | |
|---|---|---|---|---|---|
| | | | O₂min* | O₂max* | N₂O† |
| **1** Air-100% saturated | NA | NA | 293 | 322 | ∼0.01 |
| **2** Ambient | NA | NA | 1.0 | 199 | ∼0.01-0.1 |
| **3** N₂ (OFN) | 999,000 | NA | 2.2 | 6.0 | 0.00 |
| **4** N₂O | 2 | N₂ | 1.8 | 5.6 | ∼0.05 |
| **5** O₂ | 7,500 | N₂ | 12.9 | 23.2 | 0.00 |
| **6** N₂O + O₂ | 2 and 7,500 | N₂ | 12.9 | 17.3 | ∼0.05 |

NA, not applicable.
*Minimum and maximum oxygen concentrations measured for each treatment from the 12 sets of replicate experiments.
†Estimated N₂O concentration in each treatment.
The experiments were performed with water from two depths (Supplementary Table 1) at six sites (n = 12 experiments) and with six treatments (1–6). For treatment 1, the samples were sparged with compressed air and for treatment 3, with oxygen free nitrogen (OFN, 99.9%). Samples for treatments 4, 5 and 6 were sparged with each special gas as indicated. Ambient treatment 2 was simply unadulterated seawater drained straight from a Niskin into 1 litre vials. As we did not perform treatment 1 at site 1, we have a total of 70 independent measurements of the production of N₂O as a function of oxygen: 1 site × 2 Depths × 5 treatments + 5 sites × 2 Depths × 6 treatments, n = 70.

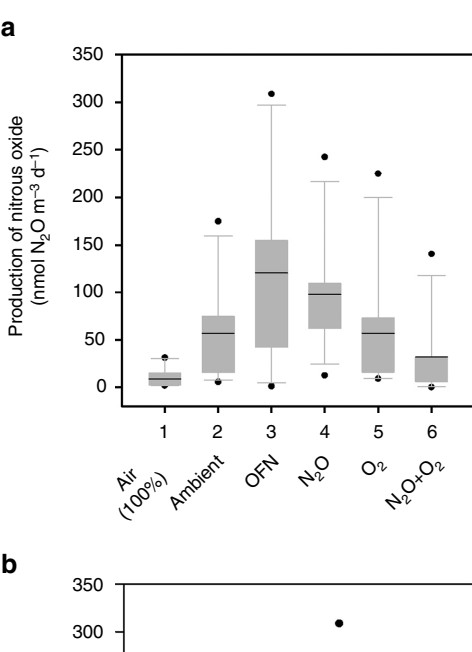

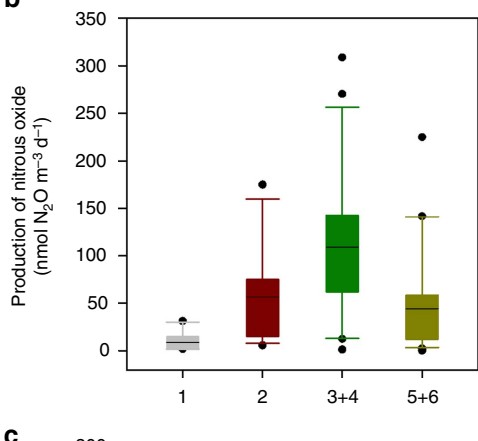

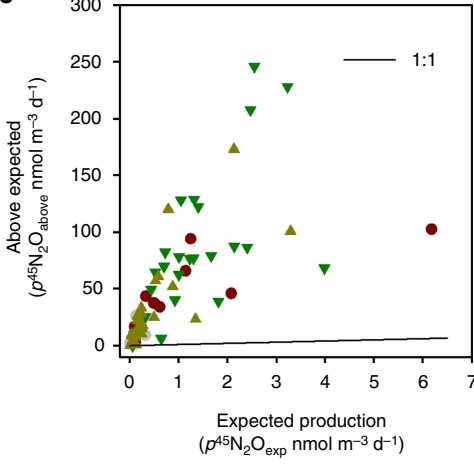

**Figure 2 | Total production of N₂O from equations (1–5) and its**
**¹⁵N-labelling.** (**a**) All oxygen treatments, including any inhibitory effect of
N₂O (50 nmol N₂O l⁻¹ crossed with O₂). (**b**) As this latter treatment had
no effect, the data were pooled by their comparative oxygen concentrations
(3 + 4 and 5 + 6; see Table 1). Each box in **a** and **b** shows the 25th and 75th
percentile, overall spread in the data and median value (horizontal line). In
both **a** and **b**, the effect of treatment is highly significant ($P < 0.0001$).
(**c**) Production of $p^{45}N_2O$ is clearly above that expected ($p^{45}N_2O_{exp}$)
from denitrification of $NO_2^-$ (see the 'Methods' section) in each treatment
and the symbol colours in **c** are the same as in **b** (grey = 100% air;
red = ambient; green = OFN and N₂O; olive-green = O₂ and N₂O + O₂).
Drawn in SigmaPlot (Systat Software, San Jose, CA, USA).

response to decreasing oxygen. The magnitude of any deviation
from this estimate, that is, the random intercept, can be used to
further explore relationships with other explanatory variables that
could potentially account for that random variation in the mag-
nitude of N₂O production. For example, there was a clear effect of
ambient oxygen concentration in the water column: with samples
collected from above 30 μmol O₂ l⁻¹ producing less N₂O, on
average, to those collected from below 30 μmol O₂ l⁻¹
(Supplementary Fig. 5a). In agreement with a growing number of
cases in the literature, we were not able to detect either of the
bacterial ammonia mono-oxygenase genes, β-*amoA* or γ-*amoA*,
but we did find high abundance of archaeal *amoA* (*AamoA*,
$5.6 \times 10^3$ copies ml⁻¹, on average). We also quantified archaeal
*nirK* (*AnirK*, $2.2 \times 10^3$ copies ml⁻¹, on average) and bacterial
*nirK* and *nirS* ($13 \times 10^3$ and $0.7 \times 10^3$ copies ml⁻¹, on average,
respectively), genes coding for the potential to reduce $NO_2^-$
(Supplementary Fig. 6). Both pairs of either archaeal or bacterial
functional genes were positively correlated with each other but
the pairs were ordinated separately in the samples collected
(Supplementary Fig. 7). There was no visual indication of a
pattern in the deviation of the random intercept and abundance
of either bacterial *nirK* or *nirS* (Supplementary Fig. 5b) but there
was a positive pattern in the abundance of *AnirK* and *AamoA*
(Supplementary Fig. 5c) that we explore further.

The fully parameterized (oxygen combined with all four
candidate genes) nonlinear mixed-effects models failed to
converge and to explore the production of N₂O as a function of
both oxygen and functional gene abundance further, we
log-transformed the data and proceeded with multiple linear
regression (Supplementary Table 3). Oxygen alone had a highly
significant negative effect on the production of N₂O (Fig. 4a,
Supplementary Table 3: M7 versus M6). Despite the compara-
tively similar abundance of bacterial *nirS* to the archaeal genes
and the greater gene abundances for *nirK*, neither *nirS* nor *nirK*
improved the fit of the model over oxygen alone, either singularly
or when combined (Supplementary Table 3: M7 versus M8, M9,
M10). Only inclusion of *AamoA* and/or *Anirk* in the model
indicated any significant influence on the overall production of
N₂O (Supplementary Table 3: M7 versus M12, M13, M14, M15).
As the model could not distinguish between the influence of
either *AamoA* or *Anirk* on the distribution of the data, we would
conclude that the most parsimonious explanation of our data is
maximal production of N₂O at lowest oxygen, combined with a
positive influence from the abundance of both archaeal functional
genes (Fig. 4). To confirm that the rate of N₂O production was
reasonable for the abundance of genes *AnirK* and *AamoA*
detected, we calculated a per copy rate (equivalent to a per cell
rate) for median N₂O production (58 nmol m⁻³ d⁻¹), below
30 μmol O₂ l⁻¹. Accordingly, 2 and 5 attomol N₂O per copy per
hour for the two genes, respectively, is representative of published
rates (2–58 attomol N₂O per cell per hour (ref. 31)).

**Depth-integrated N₂O production and sea to air exchange.** We
used the coefficients from our nonlinear mixed-effects models
(M2 and M5, Table 2) as input to a simple one-dimensional (1D)
model of N₂O, coded in R (ref. 32, see the 'Methods' section). The
objective of this was to test whether a single O₂-dependent,
N₂O production process could sustain the observed N₂O
maximum at ~100 m depth. Over a 30 day run (1 min time-
step), parameterization with M2 maintained the initial steady
state conditions (Fig. 5) without any marked accumulation
(+0.04% d⁻¹) of N₂O, whereas, with M5, we saw significant
N₂O accumulation. Note that with M2 the vast majority of the
N₂O is assumed to come from a 1:1 coupling (equations 1–5),
whereas with M5 we assume random mixing of ¹⁴NO and ¹⁵NO

**Table 2 | Output from the nonlinear mixed-effects modelling of N₂O production as an exponential function of experimentally induced decreasing oxygen.**

| Model | Parameter | Estimate | s.e. | t-value | P value | Random effect | Variance structure | AIC |
|---|---|---|---|---|---|---|---|---|
| M1 | a | 123.94 | 24.61 | 5.11 | <0.001 | Yes | No | 746 |
|  | b | 0.0535 | 0.010 | 5.12 | <0.001 | No |  |  |
| M2 | a | 120.57 | 24.17 | 4.99 | <0.001 | Yes | Yes | 730 |
|  | b | 0.0514 | 0.011 | 4.37 | <0.001 | No |  |  |
| M3 | a | 133.44 | 22.95 | 5.81 | <0.001 | Yes | Yes | 728* |
|  | b | 0.0732 | 0.015 | 4.72 | <0.001 | Yes |  |  |
| M4 | a | 137.44 | 23.09 | 5.95 | <0.001 | Yes | No | 742 |
|  | b | 0.0780 | 0.015 | 5.05 | <0.001 | Yes |  |  |
| M5 | a | 612.62 | 149.7 | 4.09 | <0.001 | Yes | Yes | NA |
|  | b | 0.0681 | 0.009 | 6.84 | <0.001 | No |  |  |

NA, not applicable.
*M3 had the lowest AIC score but its random intercept (a) and exponent (b) were highly correlated ($r = -0.99$), which suggested that the model was over parameterized and therefore M2 was taken as the most parsimonious fit to the data.
The goodness of fit for each model to the data was judged using the Akaike Information Criterion (AIC), where a lower value indicates a better fit. M2 and M3 were improved further by the addition of a power variance structure at the level of each of the 12 experiments. Note that M5 was fitted to data calculated assuming a bacterial mode of N₂O production with random isotope pairing of $^{14}NO$ and $^{15}NO$ using equations (6 and 7) ($pN_2O_{total}'$) and, as such, comparison with the other models using AIC is not appropriate. In addition, note the far higher intercept in M5, which manifests as over production of N₂O in Fig. 5b.

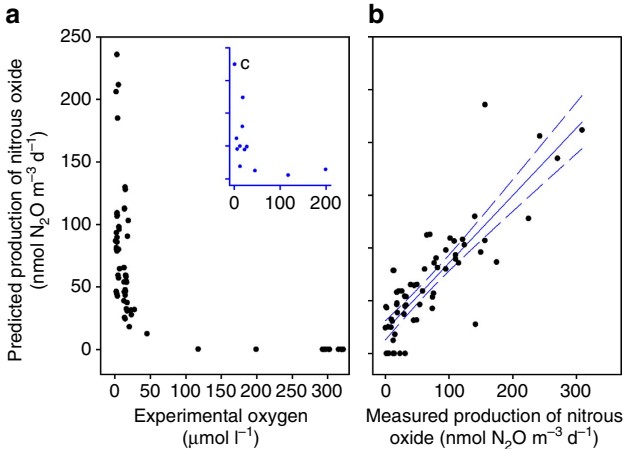

**Figure 3 | Production of total N₂O from equations (1–5) as an exponential function of decreasing O₂.** Output from the nonlinear mixed-effects model M2 (Table 2) with (**a**) predicted production of N₂O as a function of measured oxygen in each incubation bottle and (**b**) the same predicted production of N₂O as in **a**, repeated as a function of measured production (95% confidence interval). Inset (**c**) the original data for the 12 ambient, unadulterated incubations (same units as **a**). Overall, by allowing any natural variation in the production of N₂O (intercept a) to vary randomly between the 12 experiments, the nonlinear mixed-effects model captures the exponential increase in N₂O production below 30 μM oxygen well. See main text for further explanation and Supplementary Fig. 3 for the complete model output with individual fits for each experiment and overall population parameter estimates. The latter of which we then use as input to a 1D model of water column N₂O production. Drawn in SigmaPlot (Systat Software, San Jose, CA, USA).

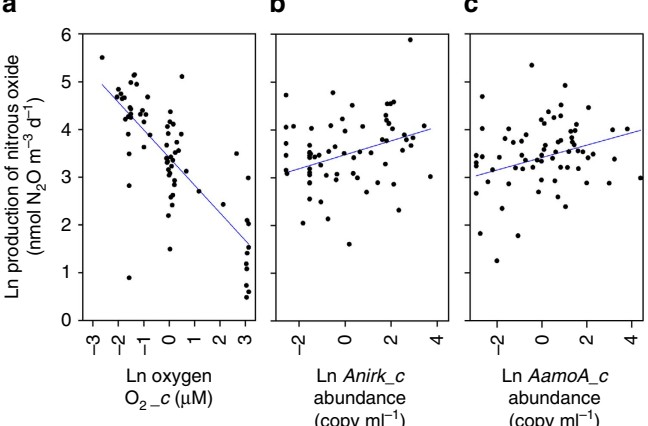

**Figure 4 | Multiple linear regression for the production of N₂O from equations (1–5) as a function of O₂ and archaeal functional gene abundance.** (**a**) N₂O production increasing with decreasing experimental oxygen (from Fig. 3a) and (**b,c**) production of N₂O increasing as a function of the abundance of both *AnirK* and *AamoA*, that is, archaeal functional gene abundance accounts for some of the random variation in N₂O production in the original nlme analysis (Table 2, Supplementary Fig. 5). The model (M14, Supplementary Table 3) suggested equal influence on the data from *AnirK* and *AamoA*, which the overwhelming $^{14}N$ and $^{15}N$ labelling of the N₂O produced corroborated further. In **a**–**c**, the x axis data have been linearized and centred, see the 'Methods' section. Drawn in SigmaPlot (Systat Software, San Jose, CA, USA).

upstream of N₂O production (equations 6 and 7). Given the better fit between our 1D model parameterized with M2, and our measured profiles of water column N₂O, we favour and proceed with M2 (see the 'Discussion' section). The sea-surface N₂O concentration in our model was fixed (Dirichlet boundary conditions) based on the average concentration from our observations (9.3 nmol N₂O l⁻¹). Sea-air exchange (efflux to the atmosphere) was therefore implicit in our model and this efflux of N₂O was sustained by a positive concentration gradient with depth, implying an upward flux of N₂O. Since the model water column was losing N₂O at the surface, it is implicit that this

upward flux of N₂O should compensate for the gas exchange loss term (that is, equal sea-air exchange). Therefore, we calculated the flux over the upper 2 and 5 m of the model water column to derive the model sea-air flux. These depths were chosen to represent the turbulent layer near the sea surface given the relatively low average wind speed (5 m s⁻¹, Supplementary Fig. 8b). The resulting sea-air flux of N₂O was 17.9 μmol N₂O m⁻² d⁻¹ and 16.0 μmol N₂O m⁻² d⁻¹ over 2 and 5 m, respectively. We compare this with an estimate from our high-resolution $pCO_2$ data ($n = 4,820$) and water column profile data ('Methods' section and Supplementary Figs 1 and 8). Accordingly, our average estimate for CO₂ exchange was 7.2 mmol CO₂ m⁻² d⁻¹ (95% confidence interval of 6.9 to 7.4), with an equivalent exchange for N₂O of 17 μmol N₂O m⁻² d⁻¹ (95% confidence interval of 15.6 to 17.5); the latter agreeing

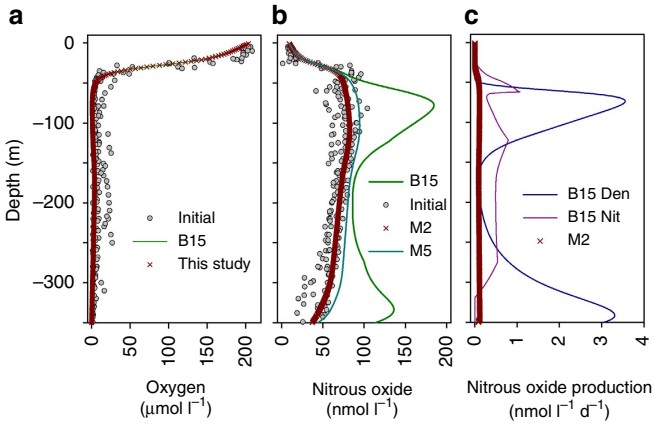

**Figure 5 | Comparison of our model output after 30 days with that of the Babbin model.** (**a**) Concentration profiles for $O_2$ and (**b**) for $N_2O$. In **b**, $N_2O$ is either produced according to B15 from Babbin et al.[5] or according to two parameterizations of equation (7), using either M2 or M5 (Table 2). In M2, we ascribe the majority of $N_2O$ production ($pN_2O_{total}$) to a 1:1 coupling, whereas, with M5, we assume that all of the $N_2O$ comes from a random mixing of $^{14}N$ and $^{15}N$ labelled NO ($pN_2O_{total}'$); note the better fit to the data with M2. (**c**) $N_2O$ production sources according to either a single response to oxygen, equation (7), parameterized with M2, or according to B15; where the two sources of nitrification and denitrification are B15 Nit and B15 Den, respectively. Initial conditions in both models were set using the mean profile from all of our observations but we show the output against all the data here to illustrate the goodness of fit of our simpler model. Drawn in SigmaPlot (Systat Software, San Jose, CA, USA).

well with the 1D model estimate for sea to air exchange of 16.0–17.9 $\mu$mol $N_2O$ m$^{-2}$ d$^{-1}$. Our experimental manipulation of oxygen, combined with a mixed-effects modelling approach has enabled us to parameterize a simple, single process 1D model that reproduces the pattern of $N_2O$ observed in the top 350 m of the tropical North Pacific.

## Discussion

Here we have shown experimentally that production of $N_2O$ increases exponentially below 30 $\mu$mol $O_2$ l$^{-1}$ and that this sensitivity to oxygen holds along a 550 km offshore transect through the OMZ of the eastern tropical North Pacific. Further, variation in the overall magnitude of $N_2O$ production correlates positively with the abundance of archaeal functional genes that potentially play a role in that production of $N_2O$. Parameterizing a simple 1D model with our experimentally derived exponential terms enabled us to model accurately the distribution of $N_2O$ over the top 350 m of the water column and, with a single response to oxygen, we could balance our estimates of sea to air exchange for $N_2O$.

Here we were following up on our previous work in the Arabian Sea[4] where the vast majority of $N_2O$ production could, apparently, be accounted for by canonical denitrification of $^{15}N$-$NO_2^-$. Accordingly, we enriched the $NO_2^-$ pool with $^{15}N$ in excess of 87 atom% and if denitrification were the dominant source of $N_2O$, and the labelling of $N_2O$ random and binomially distributed (Supplementary Table 1), then we would have expected a maximum of 23% of the resulting $N_2O$ to be single-labelled, $^{45}N_2O$. In contrast, however, we measured far more $^{45}N_2O$ than expected throughout (Fig. 2c). Put simply, the majority of N in the $N_2O$ produced was actually $^{14}N$ that was not derived from our $^{15}N$-$NO_2^-$ tracer. Although nitrite is known to have a stimulatory effect on the expression of nirK, at least in pure cultures of *Nitrosomonas europaea*, this is at nitrite concentrations of ~10 mM, far higher than that applied here

(10 $\mu$M), which is typical of $^{15}N$ tracer work for $^{15}N$-gas production[6,33,34].

One possibility is that dilution of the $^{15}N$-$NO_2^-$ pool occurred via oxidation of $^{14}N$-$NH_4^+$ to $^{14}N$-$NO_2^-$ and $^{15}N$-$NO_2^-$ to $^{15}N$-$NO_3^-$, as this would increase the chance of any subsequent denitrification producing $^{45}N_2O$, relative to $^{46}N_2O$. For this to be plausible, however, $^{14}N$-$NO_2^-$ production would need to make ~20 $\mu$mol l$^{-1}$ at the start of the incubation. We did measure significant nitrification activity (Supplementary Table 2) similar to that in the south eastern Pacific[35]. Such activity, however, could only turn over approximately 0.2% d$^{-1}$ of the $NO_2^-$ during our incubations, which would have had a negligible effect on the ratio of $^{45}N_2O$ to $^{46}N_2O$ produced. Alternatively, there could be a direct coupling between externally applied $^{15}NO_2^-$ and internally supplied $^{14}NO_2^-$, or other $^{14}N$ intermediate (for example, $^{14}NO$), from ammonia oxidation[22,23], as shown for bacterial nitrifier–denitrification[31]. Such a coupling has been argued as a possible route for $N_2O$ production in the oligotrophic North Pacific[36]. Here we were not able to detect the bacterial ammonia mono-oxygenase genes, $\beta$-*amoA* or $\gamma$-*amoA*. Given this apparent absence of any bacterial, nitrifier–denitrifier genomic potential, along with the overestimation of $N_2O$ production in our model (M5, Fig. 5b), through such a path, we would refute bacterial nitrifier–denitrification in this setting. Finally, oxidation of $NO_2^-$ has been measured simultaneously with $NO_3^-$ reduction at up to ~16 $\mu$mol $O_2$ l$^{-1}$ (ref. 35) with comparable activity of 14 nmol N l$^{-1}$ d$^{-1}$ and 21 nmol N l$^{-1}$ d$^{-1}$ for $NO_2^-$ and $NO_3^-$, respectively (median values). Although we did not quantify $NO_3^-$ reduction, if its activity were comparable to our measured rates of $NO_2^-$ oxidation it would have the same negligible effect on the $^{15}N$ labelling of the $NO_2^-$ pool.

Our oxygen experiments, combined with statistical modelling, indicate that highest $N_2O$ production is best explained by low oxygen together with a high abundance of both AnirK and AamoA. Hence, the patterns in the genomic potential for both the reduction of $NO_2^-$ (our source of $^{15}N$) and oxidation of ammonium (as a source of $^{14}N$, possibly $NH_2OH$ which has been reported as integral to archaeal ammonia oxidation[22,23]), combined with the concentration of $O_2$, account for the predominant production of $^{14}N$ and $^{15}N$ labelled $^{45}N_2O$. We could find no significant relationship for bacterial nirK and nirS. Admittedly, gene abundance (and by extrapolation cell number) does not necessarily confer a direct role on that gene for a measured process. Yet, similar positive relationships between Thaumarchaeota cell abundances and nitrification potentials are present in the low-oxygen waters of the Baltic, which, along with exponential increases in AamoA abundance below 100 $\mu$mol $O_2$ l$^{-1}$ in the Atlantic, suggest maintenance of active populations of Thaumarchaeota in low-oxygen waters[13,19].

Overall, our data agree with the growing body of evidence for archaeal-mediated $N_2O$ production[12,13]. Although the precise biochemistry of this pathway is unknown, recent reports showing a key role for $NH_2OH$ and NO in Thaumarchaeotal ammonia oxidation provide support for a 1:1 coupling in $N_2O$ production in these organisms[22,23] in support of our favoured model (Fig. 5b, M2, equations 1–5). As the nitrifying Archaea are better adapted to low oxygen compared with their bacterial analogues, it is unlikely $N_2O$ production occurs via an ammonia oxidizing bacterial type biochemical leak[37]. Rather, the metabolism of the potential precursor substrates ($NH_2OH$ and NO) in hybrid $N_2O$ formation might represent a genuine route of energy conservation[22]. Given the apparent absence of *Nor* or its equivalent in the archaea, we have to assume that our exogenous $N_2O$ (19% of total $N_2O$ production, on average) was produced through canonical denitrification operating as far as $N_2O$ but it remains to be proven whether the ability to metabolize

NO further to $N_2O$ is truly absent from archaea in the ocean. Furthermore, modelling of $N_2O$ production in the ocean[15] suggests that the oxycline is important, but has linked this $N_2O$ production primarily to bacterial ammonium oxidation (nitrification). Here we show that this is not the case and a major driver of $N_2O$ production in the ocean is likely to be archaeal hybrid $N_2O$ formation.

Although in principle, the regulatory effect of oxygen on $N_2O$ production in the ocean is widely accepted (some 100 papers citing Goreau et al.[8] in relation to ocean $N_2O$ production), its exponential form has not, to the best of our knowledge, been characterized experimentally in the ocean below $30\,\mu mol\,O_2\,l^{-1}$ (ref. 13). The basic linear $\Delta N_2O/AOU$ relationship is frequently used to model the distribution of $N_2O$ across the oceans, where the slope represents the yield of $N_2O$ per mole of $O_2$ consumed (see ref. 15 for a full discussion). This theoretical yield varies widely ($0.076$ to $0.31\,nmol\,N_2O\,\mu mol^{-1}\,O_2$ consumed) and can struggle to capture the full dynamics of $N_2O$ production in low-oxygen waters[15]. More fully parameterized versions of the $\Delta N_2O/AOU$ that allow the yield to change as a function of oxygen do a better job but still struggle at the transition (notionally $4\,\mu mol\,O_2\,l^{-1}$) from production to consumption of $N_2O$ (ref. 15). Here our mixed-effects modelling approach has enabled us to characterize a population estimate for the exponential increase in $N_2O$ production as a function of decreasing oxygen, from $1\,\mu mol\,O_2\,l^{-1}$ to $\sim 30\,\mu mol\,O_2\,l^{-1}$, without the confounding effects of individual site characteristics or, indeed, the need to invoke different metabolic pathways either side of an oxygen threshold.

The original version of the model by Babbin et al.[5] required net production of $N_2O$ from both classic nitrification and denitrification to generate the typical $100\,nmol\,l^{-1}$ peaks in $N_2O$. Here, the orginal formulation of Babbin et al., however, generated net accumulation of 20–30% $N_2O$ ($10$–$60\,nmol\,l^{-1}$ over 30 days) below the pycnocline, whereas with our single process variant the model only gained 1.2%. This suggests that the dynamics of $N_2O$ production in the two studies were fundamentally different. Indeed, we did not measure any production of $^{15}N_2$ in any of the oxygen treatments, that is, none of the $^{15}N_2O$ from the reduction of $^{15}NO_2^-$ was further reduced to $^{15}N_2$, even at $1\,\mu mol\,O_2\,l^{-1}$, which is a key feature of the Babbin model. That, along with the non-binomial distribution of $^{15}N$ in our $N_2O$, relative to the $NO_2^-$ pool, discounts denitrification as the primary source of $N_2O$ here and it is redundant in our model.

The two methods that we used to estimate sea to air exchange ($17\,\mu mol\,N_2O\,m^{-2}\,d^{-1}$, on average from the $N_2O$–$CO_2$ field data and a 1D directly parameterized model) agreed very well with each other and with those in the literature—despite different approaches. For example, $13\,\mu mol\,N_2O\,m^{-2}\,d^{-1}$ has previously been taken as representative of the ETNP[5], while a broader range of $5$–$31\,\mu mol\,m^{-2}\,d^{-1}$ has been estimated for the tropical south Pacific[2,38]. What is important for estimating the contribution from the ocean to the global $N_2O$ budget is the respective area of OMZ used for any extrapolation. The latter is partly defined by the concentration chosen for oxygen at which the microbiology either produces or consumes $N_2O$ and this is contentious[5,6]. Here we have measured a clear exponential increase in $N_2O$ production with decreasing oxygen between 1 to $30\,\mu mol\,O_2\,l^{-1}$ and apply this to regions of the ocean defined as OMZs by oxygen minima below $20\,\mu mol\,O_2\,l^{-1}$ (ref. 30). Applying our average rate to the OMZ of the ETNP ($12.4 \times 10^{12}\,m^2$ including a coastal strip making up only $\sim 3.4\%$ of the area) and the entire global extent of OMZs ($30.4 \times 10^{12}\,m^2$) generates $2.1\,Tg\,N\,y^{-1}$ and $5.1\,Tg\,N\,y^{-1}$, respectively, as $N_2O$. The latter of which agrees very well with

estimates of approximately $5.8\,Tg\,N\,y^{-1}$ derived from the oxygen-sensitive model of Nevison et al.[15]

Modelling our single response of a predominantly archaeal-driven hybrid $N_2O$ formation process not only accurately reproduces the distribution of $N_2O$ over the top $350\,m$ of the water column but this single response can also balance our high-resolution estimates of sea to air exchange for $N_2O$. Hence, a significant source of $N_2O$ that has for a long time now been ascribed to bacterial-mediated ammonium oxidation leaking $N_2O$ under oxygen stress, can better be described by an archaeal-driven hybrid $N_2O$ formation process exploiting the niche of low oxygen waters, at the margins of an OMZ.

## Methods

**Site-specific water column profiles and underway $pCO_2$ data.** A standard conductivity–temperature–depth rosette (24 Niskin (20 litres) and full Sea-Bird 24 electronics (salinity, density, $O_2$, temperature and so on) was used to collect and characterize the water at each site between 5 and 4,000 m. The distribution of $N_2O$, $CO_2$ and $NO_2^-$ was measured as described previously[4], except that the GC also had a hot-nickel catalyst and flame-ionization detector to quantify $CO_2$ after rapid equilibration and reduction to $CH_4$ (ref. 39). High temporal resolution measurements of $pCO_2$ in surface seawater and atmosphere were also made every 5 min using an underway instrument (see below).

**Production of $N_2O$ as a function of oxygen and gene abundance.** We measured the production of $^{15}N$-$N_2O$ at two depths, both within and beneath the oxycline, at each of the six sites (Supplementary Table 1). Seawater was drained from a Niskin into 4 litre Nalgene bottles and sparged for 20 min to generate six oxygen treatments (Table 1). Seawater was then dispensed under pressure into $4 \times 1$ litre clear glass moulded infusion vials (Laboratory Precision Limited), except for the Ambient treatments, which went directly into the 1 litre vials. Oxygen (50 μm calibrated electrode, Unisense) and temperature were measured and, following up on studies in the Arabian Sea[4], the vials spiked with $^{15}N$-$NO_2^-$ ([10 μM], 98 atom%, Sigma, see Supplementary Table 1 for $^{15}N$ atom % in each of the 12 sets of experiments). It is important to appreciate that all published work to date that uses $^{15}N$ to trace the production of either $N_2O$ or $N_2$ applies a 'tracer' at concentrations in excess of apparent $K_m$ values for these processes, that is, typically $5$–$10\,\mu mol\,NO_2^-\,l^{-1}$ spike, compared with $1$–$2\,\mu mol\,NO_2^-\,l^{-1}$ $K_m$ and, as such, should be considered as potentials[6,34,40]. The vials were then sealed and incubated in the dark, at $12\,°C$, for 72 h as previously[4]. Later, bacterial activity was stopped in three of the vials by the addition of 6 ml formaldehyde (36% v/v), while the fourth vial was used to measure oxygen and the water then filtered (Supor, ø = 47 mm, 0.2 μm pore size filters). The filters were immediately frozen in 2 ml cryovials, in liquid nitrogen, and stored at $-80\,°C$ for later extraction of nucleic acids (see below). Production of $^{15}N_2O$ and $^{15}N_2$ was measured in the three remaining vials against reference samples for each of the treatments, or natural abundance, by mass spectrometry (see below and Nicholls et al.[4]). The data for each triplicate were then averaged and the mean value compared with its corresponding, single measure of functional gene abundances. Genes targeted with a potential role in $N_2O$ production were: β- and γ-proteobacterial amoA; bacterial nirS, bacterial nirK, archaeal nirK, archaeal amoA (here AamoA) and, in addition, general bacterial and archaeal Marine Group I 16S rRNA genes (see Supplementary Table 4 for primer sets). A combination of the large 1 litre glass vials and multiple oxygen treatments precluded a full time series incubation in each of the 12 $N_2O$ experiments (1,400 bottles versus 280). We did, however, measure $N_2O$ production at 2, 4, 9, 18, 36 and 72 h at two sites, for two oxygen treatments during a subsequent cruise to check that our single time point incubation was not overestimating production.

**Mass spectrometry for $^{15}N_2O$ and $^{15}N_2$ and rate calculations.** All the samples were transferred under constant temperature back to the home laboratory in London and were brought to $22\,°C$ before processing. Two subsamples of the 1 litre vials were forced out under helium and transferred to either a helium-filled 12 ml gas-tight vial (Exetainer, Labco), for $^{15}N_2$ analysis, or a helium-filled 20 ml gas-tight vial (Gerstel and 20 mm butyl rubber stoppers and aluminium seals, Grace—Alltech) for $^{15}N_2O$ analysis. The 20 ml vials ended up with 10 ml of seawater and 10 ml of helium headspace to which we added a carrier of 3 nmol $N_2O$, as sparging with the compressed air, $N_2$ and $O_2$ treatments effectively removed all of the natural $N_2O$ from the samples. These were then analysed for enrichment of both single- and dual-labelled $^{45}N_2O$ and $^{46}N_2O$, respectively, against seawater samples (collected on the cruise) sparged with the five treatment gases, or, in the case of the ambient treatment, reference samples of seawater, using a trace gas pre-concentrator unit (PreCon, Thermo-Finnigan)[4]. Calibration was performed against known amounts of $N_2O$ (98 p.p.m.; BOC), and it was linear ($r^2 = 0.998$) over the range 0 to 20.72 nmol $N_2O$ absolute ($\sum^{44}N_2O$, $^{45}N_2O$ and $^{46}N_2O$).

After bringing the remaining 12 ml gas-tight vials to 22 °C, a helium headspace (1 ml) was added and the vials shaken by hand and left overnight on rollers (Spiramix) to allow $N_2$ gas to equilibrate between the water phase and headspace. Samples of the headspace (100 μl) were then analysed for enrichment in $^{15}N_2$ by injection (Multipurpose Sampler MSP2, Gerstel) into an elemental analyzer (Flash EA 1112, Thermo-Finnigan), interfaced with the continuous flow isotope ratio mass spectrometer (CF-IRMS)[4]. Calibration was performed at the beginning of each run with known amounts of oxygen free nitrogen gas (BOC) in seawater collected on the cruise, in the range of 0 to 12.6 μmol $N_2$ absolute ($\sum {}^{28}N_2$, $^{29}N_2$ and $^{30}N_2$). Values for the production of $^{29}N_2$ and/or $^{30}N_2$ were calculated as excess over the production in the time zero 'reference' samples[41].

We used $^{15}NO_2^-$ to trace the production of $N_2O$ as per our previous work in the Arabian Sea[4] and present two principle methods for calculating the total production of $N_2O$ in response to oxygen. In the first method, given that the archaea appear to lack Nor, we assume that they cannot make $N_2O$ purely from exogenous $NO_2^-$ and that any measured production of $p^{46}N_2O$ (that is, $2 \times {}^{15}NO_2^-$) must be due to canonical denitrification reducing $NO_2^-$ as far as $N_2O$. Then, any production of $N_2O$ that we cannot account for by canonical denitrification with exogenous $NO_2^-$ we assign to hybrid $N_2O$ formation, as in the most recent models for Thaumarchaeotal ammonia oxidation[22]. In the second method, we assume that all of the measured production of $N_2O$ is due to a classic bacterial-type mode of nitrifier–denitrification, with random isotope pairing of $^{14}NO$ and $^{15}NO$ upstream of the production of $N_2O$.

We calculate the overall production of $N_2O$ that we assume to be owing to canonical denitrification of exogenous $NO_2^-$ according to:

$$pN_2O_{exogenous} = p^{46}N_2O \times (FN_{NO_2^-})^{-2} \qquad (1)$$

where $FN_{NO_2-}$ is the fraction of $^{15}N$ in the $NO_2^-$ pool (Supplementary Table 1) in each set of incubations, determined by difference[34], and we ignore any turnover by either ammonium or nitrite oxidation, which is shown to be negligible relative to the size of the $NO_2^-$ pool (See the 'Discussion' section and Supplementary Table 2). We then used the measured amount of dual-labelled $p^{46}N_2O$ to predict the expected amount of single-labelled $p^{45}N_2O_{exp}$ for canonical denitrification according to[40]:

$$p^{45}N_2O_{exp} = p^{46}N_2O \times 2 \times (1 - FN_{NO_2^-}) \times (FN_{NO_2^-})^{-1} \qquad (2)$$

We would then argue that any production of $p^{45}N_2O$ above $p^{45}N_2O_{exp}$ cannot be solely due to reduction of external $NO_2^-$, and must be due to $^{15}N$ pairing with an alternative source of $^{14}N$ (for example, $^{15}NO$ from $^{15}NO_2^-$, pairing with $^{14}NH_2OH$ in archaeal hybrid $N_2O$ formation[22]) which, for simplicity, we refer to as endogenous $N_2O$:

$$p^{45}N_2O_{above} = p^{45}N_2O - p^{45}N_2O_{exp} \qquad (3)$$

$$pN_2O_{endogenous} = (FN_{NO_2^-})^{-1} \times (p^{45}N_2O + 2 \times (1 - FN_{NO_2^{-1}}) \times p^{46}N_2O. \qquad (4)$$

The first estimate of total production of $N_2O$ in our incubations with $^{15}NO_2^-$ is then the sum of the two former products:

$$pN_2O_{total} = pN_2O_{endogenous} + pN_2O_{exogenous} \qquad (5)$$

Hence, the calculation of $pN_2O_{total}$, $pN_2O_{endogenous}$ and $pN_2O_{exogenous}$ with $^{15}NO_2^-$ is synonymous to that for total $N_2$, anammox and denitrification, respectively, in all other work measuring the production of $N_2$; though the biological context is not[40]. The alternative formulation assumes that all of our measured production of $N_2O$ was dominated by a classic bacterial-type mode of nitrifier–denitrification, with random isotope pairing of $^{14}NO$ and $^{15}NO$ upstream of the production of $N_2O$ (ref. 22) and we can calculate an alternative $pN_2O_{total}'$ according to[42]:

$$pN_2O_{44} = (p^{45}N_2O / 2 \times p^{46}N_2O) \times (p^{45}N_2O + 2 \times p^{46}N_2O) \qquad (6)$$

$$pN_2O_{total'} = p^{44}N_2O + p^{45}N_2O + p^{46}N_2O \qquad (7)$$

**Molecular analysis.** In the home laboratory, each Supor filter was cut in half and one half was placed into a 2 ml sterile screw-cap tube, containing $\emptyset = 0.1$ mm glass beads. The following solutions were then added to each tube: 700 μl of 120 mmol l$^{-1}$ sodium phosphate (pH 8.0) plus 1% (w/v) acid-washed polyvinylpolypyrrolidone, 500 μl of Tris-equilibrated phenol (pH 8.0), and 50 μl of 20% (w/v) sodium dodecyl sulfate. The extraction process involved bead beating and passing the samples through hydroxyapatite and Sephadex G-75 spin columns, to separate nucleic acids from proteins and salts[43]. Nucleic acids were resuspended in 50 μl of TE (10 mmol l$^{-1}$ Tris, 1 mM EDTA [pH 8.0]) and stored at $-80$ °C.

The extracted DNA was used for quantification of functional genes (primer details are shown in Supplementary Table 4). Quantitative real-time PCR was performed in a Bio-Rad CFX96 Real-Time System. The reaction was performed in duplicate in a final volume of 15 μl, which contained 7.5 μl of SensiFAST SYBR No-ROX mix (2 ×) (Bioline), 200 nmol l$^{-1}$ of each primer and 1 μl of 10 times diluted DNA. The conditions for all reactions were as follows: 95 °C for 3 min; 40 cycles of 95 °C for 0.05 min and 60 °C for 0.30 min; 95 °C for 0.05 min; 65 °C for 0.05 min and a final step of 95 °C for 0.5 min. Absolute quantification of the targeted genes was performed with a series of 10-fold standard dilutions, using the CFX Manager version 2.0 software (Bio-Rad). Standards for bacterial 16S rRNA,

$nirS$ and $nosZ$ genes were derived from *Pseudomonas brenneri* DSM15294; environmental PCR products were used for bacterial $amoA$, $AamoA$, $nirK$, $AnirK$ and MG1 16S. Samples with Cq values that were the same or greater than those of the no template controls were assumed to be below the limit of detection (LOD). In each of these cases, the calculated LOD for the particular qPCR plate was used as the value for that sample (maximum LOD = 171 copies ml$^{-1}$). Specificity of the $AnirK$ PCR was assessed by sequencing product from a number of sites. All showed that the PCR assay was specific for its target gene (data not shown).

**Nitrification.** To account for any possible turnover of the $^{15}NO_2^-$ pool in our 72 h $^{15}N_2O$ incubations, we incubated additional water under ambient oxygen (1 to 23 μmol $O_2$ l$^{-1}$) from the second depth at each site (Supplementary Table 1). Water was sampled into 1 litre vials, allowed to overflow three times, sealed, brought to 12 °C and then, without any sparging, pushed out (2 mm Teflon tubing) under helium into the bottom of 12 ml, gas-tight vials (Exetainer, Labco), overflowed three times and sealed. The vials were then enriched from concentrated stocks (Sigma, sparged with OFN) to [10 μmol l$^{-1}$], in quadruplets, with either $^{15}NO_2^-$ or $^{15}NH_4^+$. Ammonia oxidation was estimated from the net accumulation of $^{15}NO_2^-$ after the addition of $^{15}NH_4^+$, single time point incubations (96 h); nitrite oxidation from net accumulation of $^{15}NO_3^-$ over 3, 6, 12, 24, 48 or 96 h from $^{15}NO_2^-$ and overall net nitrification from the accumulation of total $^{15}NO_x^-$ after 96 h from $^{15}NH_4^+$. The samples were fixed (50 μl 50% (w/v) ZnCl$_2$) and production of $^{15}NO_x^-$, $^{15}NO_2^-$ or $^{15}NO_x^-$ measured with a sulphamic acid assay at the University of Southern Denmark.

**Modelling $N_2O$ production.** We formulated a simple 1D model of $N_2O$ (1 m depth resolution), coded in R. The model is largely based on the parameterizations given by Babbin *et al.*[5] encompassing physical processes (upwelling, vertical diffusion and implicit gas exchange of $N_2O$) as well as biological production of $N_2O$. Vertical transport was parameterized according to Fickian diffusion with a diffusivity $K_z$ of $4 \times 10^{-4}$ m$^2$ s$^{-1}$ at the surface, decreasing linearly to $4 \times 10^{-5}$ m$^2$ s$^{-1}$ at 10 m and remained constant thereafter apart from the pycnocline (20–48 m depth) where $K_z$ was $1 \times 10^{-5}$ m$^2$ s$^{-1}$. This $K_z$ profile effectively simulated near surface turbulence while the remaining water column was dominated by diffusive processes. An upwelling velocity ($w_{up}$) of $8 \times 10^{-7}$ m s$^{-1}$ and a particle sinking velocity (of $1.2 \times 10^{-4}$ m s$^{-1}$ (ref. 5) were used.

The model resolved the upper 400 m of the water column at 1 m depth resolution and 1 min time intervals. Boundary conditions at the surface and at 400 m were fixed and prescribed by the respective averages from our profiles. This average profile also described initial conditions for $NO_3^-$, $PO_4^{3-}$, $N_2O$ and $O_2$. Particulate Organic Carbon (POC) values for the ETNP were taken from the literature[44,45], with a surface concentration of 3 μmol l$^{-1}$, a sub-surface maximum of 5 μmol l$^{-1}$ at 32 m and decreasing thereafter to 1.3 μmol l$^{-1}$ at 400 m. Model POC remineralization (POC$_{rem}$) was parameterized as a first-order process with a rate constant of $5 \times 10^{-7}$ s$^{-1}$. POC production at the surface is implicit via the fixed boundary concentration as in Babbin *et al.*[5] In addition, we parameterized POC production ($_pPOC_Z$) at depth ($Z$) as a function of the upwelling $NO_3^-$ flux and light attenuation:

$$pPOC_Z = F \times w_{up} \times [NO_3^-] \times e^{-K_d \times Z} \times r_{N:Cremin} \qquad (8)$$

where $F$ is the ratio of upwelled $NO_3^-$ used by primary producers (0.2), $[NO_3^-]$ is the concentration of $NO_3^-$ at depth $Z$, $K_d$ is the light attenuation coefficient (0.09 m$^{-1}$) and $r_{N:Cremin}$ is the N:C ratio production/remineralization ($r_{N:Cremin} = 16/106$). The value of $K_d$ was chosen as it gave a subsurface $_pPOC_Z$ maximum which was consistent with the positions of the subsurface POC- and chlorophyll-concentration maxima at the base of the mixed layer as observed during our cruise. $NO_3^-$, $PO_4^{3-}$ and $O_2$ were linked to POC production/ remineralization according to Redfield stoichiometry, as in Babbin *et al.*[5] $O_2$ consumption followed a respiratory ratio of $\sim 1.4$ ($r_{O:Crem} = 150:106$).

Production to consumption of $N_2O$ was parameterized for two separate variants of the model: (i) according to Babbin *et al.*[5] and (ii) as a function of $O_2$ concentration as described here. All processes except those producing $N_2O$ were identical in both the models. In our second variant, we parameterized model $N_2O$ production ($pN_2O$ in nmol m$^{-3}$ d$^{-1}$) using the estimates for $a$ and $b$ from our nonlinear mixed-effects models (M2 and M5, Table 2 and equation 12):

$$pN_2O = a \times e^{-b \times O_2} \qquad (9)$$

Note that the original Babbin formulation included an $[O_2]$-dependent Heaviside function which terminated $N_2O$ production when $[O_2] < 0.4$ μmol L$^{-1}$. Here, as oxygen was always above 0.4 μmol $O_2$ L$^{-1}$ it was redundant and not included in our variant of the model.

**Estimating $N_2O$ exchange using high-resolution $pCO_2$ data.** High temporal resolution measurements of $pCO_2$ in surface seawater and atmosphere were made every 5 min using an underway instrument (PML Dartcom Live $pCO_2$. UK.[46,47]) with the 'vented' equilibrator modification[46]. The equilibrator was fitted with two platinum resistance thermometers (Pico Technology, model PT100) and a water-jacket supplied with seawater from the ship's underway seawater system.

A seawater flow of 1.6 litres min$^{-1}$ was maintained through the main equilibrator. The average warming between the ship's underway seawater intake and the equilibrator was $0.2 \pm 0.1\,°C$. Atmospheric measurements of $CO_2$ were taken from an intake located on the foremast. Both gas streams from the equilibrator headspace and the air inlet were dried in a Peltier cooler ($-20\,°C$). Mixing ratios of $CO_2$ and water in the marine air and equilibrator headspace were determined by non-dispersive infrared dection (LI-840, LI-COR). Measurements were referenced against secondary calibration gases (BOC Gases, UK) with known $CO_2$ mixing ratios (257.6, 373.4 and 463.5 µmol $CO_2$ per mole) in synthetic air mixtures (21% oxygen and 79% nitrogen). All calibration gases were calibrated against certified primary standards from the National Oceanic and Atmospheric Administration (244.9 and 444.4 µmol $CO_2$ per mole). The $pCO_2$ system described here showed high consistency with a similar $pCO_2$ system and $pCO_2$ calculated from independent TA, DIC and pH during 'at sea' inter-comparison[48]. Sampling was carried out continuously (every 5 min), with the exception of periods for maintenance. See Supplementary Fig. 7 for a summary of the $pCO_2$ and wind data and resultant efflux estimates.

Then, for the samples for which high-resolution seawater $pCO_2$ ($pCO_2$sw) data were available but in which $N_2O$ was not directly quantified, we predicted molar seawater $N_2O$ concentrations ($N_2O$sw) using the linear relationship between $N_2O$sw and the molar seawater concentration of $CO_2$ ($CO_2$sw; Supplementary Fig. 1). To do this, we first estimated the $CO_2$ and $N_2O$ solubility for each sample (mol kg$^{-1}$ atm$^{-1}$; refs 49,50). $CO_2$sw for each sample was next calculated as the product of its $pCO_2$sw and corresponding molar solubility. Each resulting molar $N_2O$sw concentration was then converted to $pN_2O$sw by dividing it by the calculated, corresponding $N_2O$ molar solubility. Atmospheric $pN_2O$atm was taken as the average for samples collected from the bow of the ship throughout the ~6 week cruise ($348 \pm 6$ natm s.e., $n = 35$) and the corresponding $N_2O$ flux estimated from the high-resolution $CO_2$ fluxes calculated using the average 12 h wind speed:

$$\Delta pN_2O = pN_2O_{sw} - pN_2O_{atm} \qquad (10)$$

$$\text{Flux } N_2O = \frac{\Delta pN_2O}{\Delta pCO_2} \times \text{Flux } CO_2 \qquad (11)$$

**Statistical analyses.** All the analyses were conducted in $R$ (ref. 32) following procedures largely described in ref. 51. We began with linear mixed-effects models treating oxygen as a categorical variable and modelling $N_2O$ production as an additive, linear function of the six oxygen treatments (Table 1). With the linear mixed-effects models, we fitted the oxygen treatment as a fixed effect and included random intercepts for each of the 12 experiments, comparing models with and without 'oxygen' with likelihood ratio testing. Given that there was clear spread within the oxygen data, we then used nonlinear mixed-effects models to model $^{15}N_2O$ production as a continual, exponential function of oxygen:

$$pN_2O_{total}\left(nmol\,m^{-3}\,d^{-1}\right) = \alpha e^{(-b\,O_2\,\exp)} \qquad (12)$$

Where $pN_2O_{total}$ comes from equations (1–5) and $O_2$exp is the measured concentration of oxygen (µmol l$^{-1}$) in each incubation bottle and total production of $N_2O$ is that measured in each bottle at the end of its incubation. For the 12 sets of experiments analysed using nonlinear mixed-effects models, we either fitted both the intercept ($a$, that is, maximum $N_2O$ production) and sensitivity ($b$, that is, response to oxygen) as random effects, or, $a$ and $b$, each individually, and compared model fit in each case with the Akaike Information Criterion (AIC). Relationships between these 'random' elements, that is, variance not explained by experimental oxygen and other possible explanatory variables (for example, gene abundance) were explored visually (at the 12-experiment, group level, $n = 12$) and then more rigorously using multiple regression and the entire, linearized and centred ($x_c$) data set (natural log, $x_c = x - x_{mean}$, $n = 70$). Here, we judge the simplest model (that is, just oxygen) against more complex models (oxygen plus single or multiple functional gene abundance) also using likelihood ratio testing.

**Data availability.** The data that support the findings of this study are available from the authors on reasonable request, see author contributions for specific data sets.

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

## Acknowledgements

We thank I.A. Sanders for technical assistance and R. Nicholls and G. Yvon-Durocher for statistical guidance; the captain and crew of RRS Discovery for their support throughout the cruise (D373) and the Natural Environment Research Council (NERC) (NE/E01559X/1).

## Author contributions

M.T. and K.J.P. conceived the idea for the study, M.T. analysed the data and wrote the paper. P.-M.C. did all the cruise logistics/preparations and performed PreCon and IRMS. P.-M.C. and K.J.P. performed the molecular analyses. S.T.M. performed GC analysis on-board. V.K. remotely managed the $pCO_2$ system and with M.T. parameterized the 1D model and R.C.U.-G. scaled the $N_2O$ using the $pCO_2$. All the authors commented on and edited the manuscript.

## Additional information

**Competing financial interests:** The authors declare no competing financial interests.

**Publisher's note**: 

