## [Peer Review File · Nature Communications]

Reviewers' comments:

Reviewer #1 (Remarks to the Author):

Trimmer et al. explore potential drivers of the production of nitrous oxide [N₂O] in low-oxygen [O₂] waters of the Eastern Tropical North Pacific oxygen minimum zone. They use a combination of O₂ manipulation experiments using isotopic labeling, rate measurements, and marker gene counts to inform and parameterize a non-linear mixed effects model that shows a strong exponential relationship between declining O₂ and increasing N₂O and a significant positive correlation with counts of the Archaeal marker gene nitrite reductase (AnirK). Significant negative relationships between O₂ and N₂O concentrations have been described previously in ocean OMZ regions. Such patterns have been used to suggest a role for nitrifiers in ocean N₂O production, either directly via chemical decomposition of ammonia oxidation intermediates, or via so-called nitrifier-denitrification, in which ammonia oxidation to NO₂ is coupled to NO₂ reduction by the same organism. Here, through some of the first experiments to test the N₂O and O₂ relations, the authors observe an excess of single ¹⁵N-labeled N₂O in incubations with ¹⁵NO₂ coupled with a strong positive relationship between AnirK abundances and N₂O production rates, and use these results to implicate Archaeal nitrifier-denitrification as the most likely source of N₂O. Overall, this is a comprehensive, statistically robust, and well-conceived study that provides strong support for the hypothesis that ammonia-oxidizing Archaea play an important role in N₂O production. Broadly, these results will be useful for helping constrain models to determine effects of O₂ content on greenhouse gas cycling.

There are, however, several issues that deserve attention prior to publication.

1) Gene/cell counts data not well described, and questionable. Why are AnirK counts 2 orders of magnitude higher than those of AamoA (~10⁶ vs 10⁴ per ml) if both genes are putatively localized in the same organism)? Indeed, Figure 4 shows maximum AnirK counts of ~5,000,000 per ml, with an average closer to about 200,000 per ml (~ln 12). Frankly, these values seem high to me. Is 10⁶ AnirK per ml consistent with AOA abundance in this system (or in comparison to other systems; e.g., ETSP), and with knowledge of nirK copy number in available Thaumarchaeota genomes? Of course, the magnitude of difference between AnirK vs AamoA counts cannot be explained by copy number variation alone. Is it possible that the AnirK primers are non-specific and that the qPCR assays are also amplifying denitrifier nirK? Or something else? (Was there any attempt to confirm the specificity of these primer sets for this study?) Some (rough) insight into these questions could presumably come from the counts of total Marine Group I (MGI) 16S copies (Table S5), although these data are not presented (why???), or by considering prior literature on correlations among MGI 16S, AamoA, and AnirK genes (e.g., e.g., Lund et al. 2012, ISME find a much smaller difference in AamoA vs AnirK counts). These questions are important for validating the integrity of AnirK as a marker for Archaeal ammonia oxidation.

2) Per-cell rates? A more transparent discussion of the gene count data could be used to help bound per-cell estimates of N₂O production (based on the experimental incubations). Doing so would be useful, at the very least for determining whether the observed rates, if they are not realistic at the per-cell level based on knowledge from the literature, could be driven partly by other processes.

3) N₂O production mechanism. It remains unclear how AOA generate N₂O from NO₂. Many AOA genomes, including several from marine environments, encode only nirK, and those AOA genomes that do encode components of downstream denitrification steps are missing key catalytic subunits (e.g., of nitric oxide reductase). Given this lack of mechanistic understanding, the extent to which AnirK may a priori be considered a reliable proxy for AOA-based N₂O production is (seemingly) debatable. The choice of AnirK as a marker, and the state of knowledge about the NO₂-to-N₂O production step should be briefly discussed (building upon the brief mention in line 239).

4) "Experimental component" could be better articulated/emphasized in the Intro. It took a second reading to realize that the paper was trying to draw attention (e.g., in para 2 of the Intro, para 1 of the Discussion) to the fact that this work is one of the first (the first?) studies to show the inverse relation between N₂O and O₂ using "experiments", rather than environmental data. I know that this is stated in the Intro, but it is done in such a way as to be easily overlooked (it happened in my first reading of the ms, so it is possible for others as well). To better highlight this novel aspect of the work, I suggest moving some of the stronger statements from lines 246-249 into the Intro. This will immediately establish a contrast to prior work.

5) Nitrite/ammonia oxidation rates. It would be useful to provide a brief mention of how the measured rates compare (consistent/inconsistent?) to those previously observed in the eastern Pacific (e.g., Beman et al. 2012, Kalvelage et al. 2013, Ganesh et al. 2015, etc).

6) Treatments/levels need more explanation. Notably, the treatment names "N₂, N₂O, N₂O + O₂, etc" (see Table 1) are not well described. As a consequence, the first mention of "12 experiments" (line 109) is somewhat baffling. Please briefly clarify (in the Results) the overall experimental plan.

7) Target depths. Following on comment #6, the main text does not actually specify the two target depth zones from which samples were collected. Indeed, this information is cryptic even in the Methods at the end of the manuscript. Please clarify.

8) Chemical concentrations/context. The Intro would benefit from additional details regarding the environmental conditions/significance of the target study area. How much N₂O production is actually associated with OMZs ("significant sources" in line 30 is vague, and not further qualified)? From prior work, how much N₂O is present? What are the "representative oxygen concentrations" referred to in line 76?

9) "Bottle effects" (lines 123-12129) discussion unclear. Bottle effects can mean a variety of things. Please clarify the exact bias or pattern being tested for here, and how time series sampling is helping rule out such effects.

10) qPCR standard curves. I did not see a description of how the standard curve for the qPCR assays was generated. Specifically, what was the source of standard template DNA?

Other:

Table 1 and line 401. Define "OFN".

Line 108. Should "dual labeled" instead be "single labeled"? (based on data elsewhere in paper; e.g., lines 204-207)

Line 125. What is the "main" experiment?

Reviewer #2 (Remarks to the Author):

Review MS NCOMMS-16-03439

General Comment:

Trimmer et al. present results on nitrous oxide (N₂O) production as a function of dissolved oxygen availability in the 1 to 30 μm range and archaeal gene abundance in the North Pacific. This is a careful study that focuses on the mechanistic basis for production of nitrous oxide in low-oxygen

marine environments. This is an important and unresolved problem in the marine nitrogen cycle research and of potential interest to a broad audience. Overall, the manuscript is well written and organized. Field based measurements of nitrous oxide concentrations, isotopic signatures, and substrates utilized to facilitate its production are relatively scarce in oceanic environments, and the data are valuable.

However, the interpretation of the mechanism used to produce nitrous oxide by ammonia-oxidizing Thaumarchaea needs to be re-assessed, considering the recent findings that incorporate the production and evolution of "nitric oxide (NO)" by ammonia oxidizing archaea (see below).

Specific Comments:

1. Two recent publications (Martens-Habbena et al. 2015; Kozlowski et al. 2016) using pure cultures of marine and terrestrial Thaumarchaea (*Nitrosopumilus maritimus* SCM1 and *Nitrososphaera viensis*) have provided evidence for "nitric oxide" as an important intermediate produced by both organisms. The production of nitrous oxide by ammonia-oxidizing Thaumarchaea was conceptualized based on a side reaction involving nitric oxide produced by the ammonia-oxidizing organism that was subsequently converted to nitrous oxide by "a reaction with water". While the most convincing of these studies utilized a soil isolate, the central biochemistry is conserved among both isolates, and genes and transcripts associated with these reactions are often detected in metagenomic and metatranscriptomic surveys. The essential inhibition of nitrification in isolates and coastal marine waters with a nitric oxide scavenger, PTIO, provides further support for this mechanism.

2. While gene abundances were used in this study to explain the nitrous oxide signals, utilization of transcript would be more convincing, and perhaps yield more explanatory power over genes abundances alone. This is especially true for marine Thaumarchaea, which often display high activities at the base of the oxycline, but are often at abundance levels that are considerably lower. The metatranscriptomic analysis of the eastern tropical South Pacific oxygen minimum zone by Stewart et al. (2012) is an example that displays the mismatch between gene abundance and activity quite well.

3. The choice of oligo-nucleotide primers used to assess the abundance of the *amoA* gene corresponding to both Bacterial and Thaumarchaeal groups is a bit surprising, given that both are utilized primarily in terrestrial environments, and the later is considerably degenerate. The primers developed by Mincer et. al. 2007 to target the Thaumarchaea are most commonly used. What was the specific justification for utilizing the primers used by Tourna et. al.?

4. (Minor) L128 "The overwhelming majority of studies have argued for nitrification by ammonia oxidizing bacteria ..." This is a weak and old argument by now.

References:

Kozlowski et al. (2016). Pathways and key intermediates required for obligate aerobic ammonia-dependent chemolithotrophy in bacteria and Thaumarchaeota ISME J. advanced online publication. Feb 16. doi: 10.1038/ismej.2016.2.

Martens-Habbena et al. (2015). The production of nitric oxide by marine ammonia-oxidizing archaea and inhibition of archaeal ammonia oxidation by a nitric oxide scavenger. *Environ Microbiol.* 17: 2261-74.

Mincer et al. (2007) Quantitative distribution of presumptive archaeal and bacterial nitrifiers in Monterey Bay and the North Pacific Subtropical Gyre. *Environ Microbiol* 9: 1162-1175.

Stewart et al. (2012). Microbial metatranscriptomics in a permanent marine oxygen minimum

zone. *Environ Microbiol.* 14: 23-40.

Reviewer #3 (Remarks to the Author):

The paper reports an extensive biogeochemical and microbiological study of rates and pathways of N₂O formation in the world's largest oxygen minimum zone (OMZ), concluding that N₂O production can be ascribed to ammonium-oxidizing Archaea through coupled nitrification-denitrification, and that rates can be predicted from a simple oxygen dependent rate expression. OMZs are recognized as important sources of atmospheric N₂O, but there is no consensus on the mechanisms and regulation of N₂O production there. Thus, if the conclusions hold, the study represents a substantial step forward, which should be of interest to a wide audience.

The strengths of the study include the high spatial coverage and large experimental dataset of good quality. As stated in the paper, it is the first detailed experimental investigation linking N₂O production under in situ conditions to a specific microbial pathway. In its present form, however, a number of issues appear to undermine the central conclusions.

1) The reported ¹⁵N-N₂O production rates greatly underestimate total N₂O production. N₂O production rates are based on incubations with ¹⁵N-labeled nitrite, a central finding being that the ¹⁵N-labeled N₂O formed is mainly of mass 45 (¹⁵N¹⁴NO) rather than mass 46 (¹⁵N¹⁵NO) as would be predicted from the isotopic composition of nitrite. This leads to the (correct) conclusion that N₂O is mainly stems from another source than nitrite and (l. 208) that the majority of N in N₂O is actually ¹⁴N - which implies that most of the N₂O produced during the experiments accumulated as ⁴⁴N₂O (¹⁴N¹⁴NO). Based on the relative accumulation of masses 45 and 46 (Fig. 2c), typically {greater than or equal to} 50:1 but highly variable, and assuming random isotope pairing during the denitrification step to N₂O, we can estimate that total ⁴⁴N₂O production rates in the experiments were typically at least 10 times higher than ¹⁵N-N₂O and that this factor varied strongly between individual experiments (the situation is analogous to the use of the isotope pairing technique to quantify denitrification in aquatic sediments where a large fraction of N₂O/N₂ is formed through coupled nitrification-denitrification). This has serious implications: Firstly, the correlations of N₂O production to oxygen, gene numbers, etc., must apply to the total rates (or ⁴⁴N₂O rates) and not only to the ¹⁵N-N₂O rates as demonstrated now. It is not clear if this will be the case, because each data point should be scaled individually depending on the ⁴⁵N₂O/⁴⁶N₂O production ratio. Secondly, the rate expression used for modelling should also be based on total rates. As far as I can see, increasing the rates by an order of magnitude will lead to a similar increase in the modelled N₂O concentration (Fig. 5), which means that the model will no longer fit the measurements. The experimental dataset needs to be re-evaluated.

Furthermore, it should be acknowledged that the experiments are, presumably, blind to "leaky" nitrifier N₂O production through the hydroxylamine pathway. Thus, total N₂O production could be even larger than what the calculations mentioned above will show. This could have been investigated by measurement of N₂O production in the ¹⁵N ammonium incubations, which were included in the study. Did the authors attempt such measurements?

2) The archaeal nirK gene numbers appear orders of magnitude too high. The central conclusion that N₂O is produced through archaeal nitrification-denitrification rests strongly on the correlation of rates and AnirK gene numbers determined by qPCR. These numbers are extremely high (Fig. S5), up to 10⁶ mL⁻¹, which is similar to or higher than total prokaryote counts in mesopelagic waters, and 2 orders of magnitude higher than the counts of archaeal amoA genes in the same samples. While the cells may have more than one copy of each gene, it is unlikely that they have 100 copies of nirK. Furthermore, a previous study (Lund et al. 2012) found a reasonable agreement between AnirK and AamoA in North Pacific waters, and typical numbers of 10⁴ mL⁻¹, in agreement with the AamoA counts here and in another study from the same OMZ (Beman et al.

2012, L&O). This discrepancy might suggest that the nirK primer set as applied here is not specific for thaumarchaeotal ammonium oxidizers, which clearly undermines the conclusion based on the correlation.

3) The background of the paper, as presented here, is a bit of a strawman. The authors argue that the present understanding is that N₂O formation in OMZs is due to bacterial ammonium oxidizers. This is an out-dated point of view, which originates from the days before the role of Thaumarchaeota was realised. Studies by Francis, Beman, Stewart and others have clearly documented that bacteria play a very minor role in the ammonium oxidizer community and in ammonium oxidation in OMZs, and Santoro and coworkers have shown that Thaumarchaeota are a likely source of N₂O there, and that at least part of their N₂O production is through nitrification-denitrification. The novelty of the present story lies in the experimental approach and derivation of quantitative relationships, not in discovering the role of Thaumarchaeota.

Specific comments

19: through 550 km is not informative specify inshore-offshore gradient or similar

21: strong 45N₂O signature makes no sense without explanation of the methods

46: here and in many other places the authors use quotation marks in an unclear manner - does this indicate dubious results or what?

48: explain "N₂O anomaly"

64: Thaumarchaeota is used elsewhere and should be correct

87: unclear sentence - how can the MLD extend to 25 m and then increase steadily?

92: what is "the true anoxic core"??

93: unclear: nitrite and N₂O minima were found at 350 m and occurred at 400-450 m??

Fig. S2 is very difficult to understand. Why are the same data points shown in multiple frames in both a and b? Why do the sections overlap sometimes but not always?

107: is there such a thing as O₂-free nitrogen gas? (and OFM is used but not defined anywhere)

108: dual labelled 45N₂O??

120: exponential increase is confusing without stating the direction of oxygen concentrations - it is an exponential decrease with increasing oxygen

121: it is not obvious to me that the exponential function is consistent with the in situ distribution

124: these developments are by no means "approximately linear". Two out of four clearly cease completely after 12 h, and the two others clearly decelerate after 20 h - the effect seems to be depth dependent. The underestimation of rates in the long incubations adds further uncertainty to issue #1 above.

200-2: please explain - why does the fact that nitrification is active lead the authors to expect no N₂O production from nitrite? Nitrifier denitrification depends on nitrification.

209-11: the relevance of this sentence is not clear.

225-6: the authors did not measure net nitrification (i.e. increase in concentrations of nitrite+nitrate), I believe. Furthermore, Kalvelage and coworkers have reported nitrate reduction to 20 μM oxygen in OMZ waters, so why would 1 μM exclude the process? The potential for nitrate being the source of 14N in N₂O should be discussed further.

228-9: what does "and, though not directly" mean?

240-1: this is pure speculation - being better adapted doesn't mean being perfect

260: Babbin did not present a variant of the authors' model - it is the other way around

276-8: this seems to contradict the general applicability of the simple model advocated earlier

281-2: air-sea exchange rates should be independent of the assumptions about pathways of N₂O production?

320: how were vials filled with He is they were only sealed later?

442: where does the Heaviside function come from? The data implies increasing N₂O production all the way to zero oxygen. What does a* refer to - it is not in the equation? (here again, why the quotation marks?)

Fig. 1: Specify definitions of the OMZ boundary and N deficit. It would be useful to have different markers for the different stations in c) and d).

Fig. 2: Why are the highest values in panel b) not included in c)? E.g. two high outliers in treatments 3+4, one in treatment 5+6.

Fig. 4: aomA should be amoA

Table S2: The explanation of the rates is highly confusing. According to methods, nitrite oxidation was apparently from incubations with ^{15}N nitrite. If nitrification is the sum of ammonium and nitrite oxidation, how can the values be lower than nitrite oxidation?

Table S5: Where are the results from the quantifications of non-candidate genes (bacterial 16S RNA etc.)? What is the justification for the choice of AnirK primers

All 3 reviewers highlighted issues with the qPCR data. We reviewed the data and the methods used to collect it. In doing so it quickly became apparent that the PDRA who did the analyses had made some significant, and somewhat bizarre errors in her determination of the gene copies in the standards used in these qPCR assays. These included errors associated with poor DNA quantification but also a failure by the PDRA to do simple calculations properly. Given these issues we have gone back to the raw data and recalculated everything from scratch again, ensuring these were correct this time. This has altered all the counts and produced much lower counts for general bacterial 16S and AnirK such that there is now a much closer correlation between AamoA and AnirK counts (ratio AnirK:AamoA mean = 1.98, median = 0.41) and far better agreement with previous counts of these genes in other aquatic environments. We have also sequenced AnirK PCR products from these samples and they are overwhelmingly most closely related to other AnirK sequences, which we have now reported in the manuscript. Therefore we are confident that these data are now correct, that our analysis is specific for AnirK and so the questions raised by both Reviewer 1 and 3 about possible amplification of non-AnirK genes we think are no longer a concern.

Reviewer 1 also suggested that AnirK may not be an effective marker for ammonia oxidation. AnirK is a marker for nitrite reduction leading to the production of N₂O NOT for ammonia oxidation and our sequence data validates the PCR we have used as an effective method to analyse this gene. However, the most important point is that the abundance of this gene, and AamoA, is correlated with N₂O production. These correlations support our suggestion of AOA nitrifier-denitrification as the source of N₂O as our work has shown it cannot be produced by canonical denitrification or, indeed, bacterial ammonium oxidation.

Reviewer 2 suggests that using transcript analysis may improve our linkage between the activity of AOA and N₂O production. However, analysis of mRNA from natural samples is very challenging and the relationship between transcript copy number, translated protein copies and activity is very difficult to define effectively for model organisms in the laboratory, let alone environmental organisms in situ (see. Pedneault et al. 2014 Scientific Reports 4 4661; Taniguchi et al. 2010 Science 329 533). Gene abundance is an effective proxy for cell numbers and does show correlation to activity in our analysis, which is exactly what would be expected and therefore we cannot see how an RNA-based analysis is likely to clarify the message we are presenting. The reference to Stewart et al. 2012 seems odd to us as this was a purely molecular analysis without any process measurements at all. Whilst it shows that gene and transcript abundance are variable (across many genes) it cannot show which is a better measure of an organism's contribution to an actual process in situ.

Point by point to each reviewer.

Reviewer #1 (Remarks to the Author):

Trimmer et al. explore potential drivers of the production of nitrous oxide [N₂O] in low-oxygen [O₂] waters of the Eastern Tropical North Pacific oxygen minimum zone. They use a combination of O₂ manipulation experiments using isotopic labeling, rate measurements, and marker gene counts to inform and parameterize a non-linear mixed effects model that shows a strong exponential relationship between declining O₂ and increasing N₂O and a significant positive correlation with counts of the Archaeal marker gene nitrite reductase (AnirK). Significant negative relationships between O₂ and N₂O concentrations have been described previously in ocean OMZ regions. Such patterns have been used to suggest a role for nitrifiers in ocean N₂O production, either directly via chemical decomposition of ammonia oxidation intermediates, or via so-called nitrifier-denitrification, in which ammonia oxidation to NO₂ is coupled to NO₂ reduction by the same organism. Here, through some of the first experiments to test the N₂O and O₂ relations, the authors observe an excess of single ¹⁵N-labeled N₂O in incubations with ¹⁵NO₂ coupled with a strong positive relationship between AnirK abundances and N₂O production rates, and use these results to implicate Archaeal nitrifier-denitrification as the most likely source of N₂O. Overall, this is a comprehensive, statistically robust, and well-conceived study that provides strong support for the hypothesis that ammonia-oxidizing Archaea play an important role in N₂O production. Broadly, these results will be useful for helping constrain models to determine effects of O₂ content on greenhouse gas cycling.

There are, however, several issues that deserve attention prior to publication.

- 1) Gene/cell counts data not well described, and questionable.

Why are AnirK counts 2 orders of magnitude higher than those of AamoA (~10⁶ vs 10⁴ per ml) if both genes are putatively localized in the same organism)? Indeed, Figure 4 shows maximum AnirK counts of ~5,000,000 per ml, with an average closer to about 200,000 per ml (~ln 12).

Frankly, these values seem high to me. Is 10⁶ AnirK per ml consistent with AOA abundance in this system (or in comparison to other systems; e.g., ETSP), and with knowledge of nirK copy number in available Thaumarchaeota genomes?

Please see the response to qPCR and molecular analysis above

Of course, the magnitude of difference between AnirK vs AamoA counts cannot be explained by copy number variation alone.

Is it possible that the AnirK primers are non-specific and that the qPCR assays are also amplifying denitrifier nirK? Or something else?

Please see the response to qPCR and molecular analysis above

(Was there any attempt to confirm the specificity of these primer sets for this study?)

Some (rough) insight into these questions could presumably come from the counts of total Marine Group I (MGI) 16S copies (Table S5), although these data are not presented (why??), or by considering prior literature on correlations among MGI 16S, AamoA, and AnirK genes (e.g., e.g., Lund et al. 2012, ISME find a much smaller difference in AamoA vs AnirK counts).

The MGI and general bacteria (mentioned by Reviewer 3) data are not used at all in the manuscript because they add nothing to our analysis or understanding of the production of N₂O in these waters. However, with our reanalyzed qPCR data these data are broadly similar to AamoA and AnirK counts in these samples and to illustrate this we have added a comparative figure to the Supplementary Information (Fig S6) that is cited in the results. Line 158.

These questions are important for validating the integrity of AnirK as a marker for Archaeal ammonia oxidation.

Please see the response to qPCR and molecular analysis above

- 2) Per-cell rates? A more transparent discussion of the gene count data could be used to help bound per-cell estimates of N₂O production (based on the experimental incubations). Doing so would be useful, at the very least for determining whether the observed rates, if they are not realistic at the per-cell level based on knowledge from the literature, could be driven partly by other processes.

Even though this issue is probably linked to the errors in our original count data it is a good suggestion and would confirm that our data lie with a range that is supported by previous work. The per cell rate for the N₂O production rates we measure at oxygen concentrations below 30 μM is 2 or 5 attomol N₂O per copy (cell) h⁻¹ for AnirK and AamoA respectively. Shaw et al. (Environ. Microbiol. 2006) report 2-58 attomol N₂O per cell h⁻¹ for Nitrosospira. We have added a couple of lines into the text to report this and there is no need to invoke “other processes” as suggested by the reviewer. Lines 175-179.

- 3) N₂O production mechanism. It remains unclear how AOA generate N₂O from NO₂. Many AOA genomes, including several from marine environments, encode only nirK, and those AOA genomes that do encode components of downstream denitrification steps are missing key catalytic subunits (e.g., of nitric oxide reductase). Given this lack of mechanistic understanding, the extent to which AnirK may a priori be considered a reliable proxy for AOA-based N₂O production is (seemingly) debatable. The choice of AnirK as a marker, and the state of knowledge about the NO₂-to-N₂O production step should be briefly discussed (building upon the brief mention in line 239).

It is true that it is unclear how AOA generate N₂O from nitrite, but with the presence of AnirK it is clear these organisms can reduce nitrite, presumably to NO. The failure to detect other denitrification genes has to be seen in the context of just how different archaeal genes are from bacterial homologs. AamoA is really quite different from bacterial amoA so the issue here (no clear nitric oxide reductase-type gene) may well be due to gene divergence between the archaeal and bacterial clades. What is clear is that N₂O production from AOA is common, and supported as a process in situ by Santoro’s work and from a molecular standpoint by Lund et al., whose primers we have used. Thus, AnirK has good provenance in the literature as a marker for AOA N₂O production. The Kozlowski paper mentioned by Reviewer 2 below does suggest that the pathway to N₂O in AOA may be distinct from a classic bacterial nitrifier-denitrification route. However, that does not alter the story we present here nor the data we present in

Figs 3, 4 and 5.

4) "Experimental component" could be better articulated/emphasized in the Intro. It took a second reading to realize that the paper was trying to draw attention (e.g., in para 2 of the Intro, para 1 of the Discussion) to the fact that this work is one of the first (the first?) studies to show the inverse relation between N₂O and O₂ using "experiments", rather than environmental data. I know that this is stated in the Intro, but it is done in such a way as to be easily overlooked (it happened in my first reading of the ms, so it is possible for others as well). To better highlight this novel aspect of the work, I suggest moving some of the stronger statements from lines 246-249 into the Intro. This will immediately establish a contrast to prior work.

That's a great idea, thanks. See new and edited lines 42 to 61.

5) Nitrite/ammonia oxidation rates. It would be useful to provide a brief mention of how the measured rates compare (consistent/inconsistent?) to those previously observed in the eastern Pacific (e.g., Beman et al. 2012, Kalvelage et al. 2013, Ganesh et al. 2015, etc).

The comparison with measurements by Kalvelage 2013 is made on line 229; yes it is very brief but supplemental measurements of nitrification are not our focus and we are tightly constrained by space.

6) Treatments/levels need more explanation. Notably, the treatment names "N₂, N₂O, N₂O + O₂, etc" (see Table 1) are not well described. As a consequence, the first mention of "12 experiments" (line 109) is somewhat baffling. Please briefly clarify (in the Results) the overall experimental plan.

Yes this was inadequate. We now describe our experimental design briefly at the start of the results section -Nitrous oxide production as a function of oxygen- See lines 106-110 and have fully revised the description of the treatments in Table 1.

7) Target depths. Following on comment #6, the main text does not actually specify the two target depth zones from which samples were collected. Indeed, this information is cryptic even in the Methods at the end of the manuscript. Please clarify.

We have clarified this as part of point 6 above. Our aim was to generate natural variation in both ambient oxygen and nitrous oxide concentrations which we largely achieved.

8) Chemical concentrations/context. The Intro would benefit from additional details regarding the environmental conditions/significance of the target study area. How much N₂O production is actually associated with OMZs ("significant sources" in line 30 is vague, and not further qualified)? From prior work, how much N₂O is present? What are the "representative oxygen concentrations" referred to in line 76?

The first thing to say of course is that the amount of N₂O production associated with OMZs remains rather uncertain and maybe why most papers tend to shy away from explicitly assigning a value, instead opting for vague terms such as "major" or "substantial". We have included a reference and given a range for global OMZs N₂O emissions of 0.8-1.35 Tg N yr⁻¹ compared to an oceanic total of 1.8-5.8, i.e. OMZs roughly in the range 20-75% of the oceanic N₂O total (and excluding coastal ecosystems). We have also included a value for oxygen under which N₂O shows strong accumulation (<62.5 μmol l⁻¹) where N₂O is known to accumulate, keeping it as brief as possible in both cases, see lines 30 and 78.

9) "Bottle effects" (lines 123-12129) discussion unclear. Bottle effects can mean a variety of things. Please clarify the exact bias or pattern being tested for here, and how time series sampling is helping rule out such effects.

Yes it was unclear. We merely wanted to check if our 72h incubation overestimated production, as some related work with $^{15}\text{N}-\text{N}_2$ has suggested that it might. Work with ^{15}N for N_2 production by anammox or denitrification uses regression of production over time but these studies are usually only interested with measuring N_2 under ambient conditions and reporting those observations. Here we wanted to experimentally test the effect of oxygen on N_2O but a combination of the large 1L glass-vials and multiple oxygen treatments precluded a full time series incubation in each of the 12 main experiments with N_2O by oxygen treatment i.e. 1400 1L bottles versus our 280. We did measure its production at 2, 4, 9, 18, 36 and 72h at two oxygen saturations and for two depths to check that production was approximately linear (Fig. S4). Clearly production wanes after 18 h but where it was strongest and most linear (90m) the rates recorded $\sim 30 \text{ nmol m}^{-3} \text{ d}^{-1}$ and $56 \text{ nmol m}^{-3} \text{ d}^{-1}$ are representative of our overall median value of $58 \text{ nmol m}^{-3} \text{ d}^{-1}$. At 60 m production was weak and markedly non-linear over 72h but not representative of our main data set at $\sim 8 \text{ nmol m}^{-3} \text{ d}^{-1}$. Note, that the total amount of N_2O varied significantly across the 12 main experiments i.e. the deviation in the random intercept in Fig. S5 but this is captured by the mixed effects model which does a good job at parameterising the 1D model (Fig 3 and 4, main text). See lines 132 to 137.

10) qPCR standard curves. I did not see a description of how the standard curve for the qPCR assays was generated. Specifically, what was the source of standard template DNA?

The trouble our PDRA had with the qPCR data notwithstanding, this method is now such a standard technique in the field we did not feel a detailed description was necessary here. However we have amended the text to indicate the source of the standards for each gene. See Methods lines 407 to 411.

Other:

Table 1 and line 401. Define "OFN".

Done. See lines 738-739, Table 1.

Line 108. Should "dual labeled" instead be "single labeled"? (based on data elsewhere in paper; e.g., lines 204-207)

People see this in different ways – dual can be taken to mean one of each 14 and 15 to give $^{45}\text{N}_2\text{O}$, with single being taken as pure ^{15}N to make $^{46}\text{N}_2\text{O}$. We have changed it to single and define it as such in the methods and main text. Lines 115, 220, 351, 367.

Line 125. What is the "main" experiment?

Redundant and removed.

Reviewer #2 (Remarks to the Author):

General Comment:

Trimmer et al. present results on nitrous oxide (N₂O) production as a function of dissolved oxygen availability in the 1 to 30 μm range and archaeal gene abundance in the North Pacific. This is a careful study that focuses on the mechanistic basis for production of nitrous oxide in low-oxygen marine environments. This is an important and unresolved problem in the marine nitrogen cycle research and of potential interest to a broad audience. Overall, the manuscript is well written and organized. Field based measurements of nitrous oxide concentrations, isotopic signatures, and substrates utilized to facilitate its production are relatively scarce in oceanic environments, and the data are valuable.

However, the interpretation of the mechanism used to produce nitrous oxide by ammonia-oxidizing Thaumarchaea needs to be re-assessed, considering the recent findings that incorporate the production and evolution of "nitric oxide (NO)" by ammonia oxidizing archaea (see below).

Specific Comments:

1. Two recent publications (Martens-Habbena et al. 2015; Kozłowski et al. 2016) using pure cultures of marine and terrestrial Thaumarchaea (*Nitrosopumilus maritimus* SCM1 and *Nitrososphaera viensis*) have provided evidence for "nitric oxide" as an important intermediate produced by both organisms. The production of nitrous oxide by ammonia-oxidizing Thaumarchaea was conceptualized based on a side reaction involving nitric oxide produced by the ammonia-oxidizing organism that was subsequently converted to nitrous oxide by "a reaction with water". While the most convincing of these studies utilized a soil isolate, the central biochemistry is conserved among both isolates, and genes and transcripts associated with these reactions are often detected in metagenomic and metatranscriptomic surveys. The essential inhibition of nitrification in isolates and coastal marine waters with a nitric oxide scavenger, PTIO, provides further support for this mechanism.

We are a little confused by this point and hope we can clarify. As stated above the actual pathway of N₂O production in AOA is still not clear but both of the mentioned papers support a role for NO in Thaumarchaeal metabolism, which in no way negates anything we say. In fact, NO is implicit in our argument that N₂O production is in part explained by AnirK abundance, as we would expect this archaeal analogue of nirK to code for a nitrite reductase whose product would indeed be NO. Here, variation in the abundance of AnirK and AamoA correlates with N₂O production. We added ¹⁵NO₂⁻ but mainly produced ¹⁴N and ¹⁵N labelled ⁴⁵N₂O that cannot be due to nitrite in the water (see R3 for further details). We then discussed the numerous options that could lead to this, coming down firmly on the assumption of "a direct coupling between externally applied ¹⁵NO₂⁻ and internally supplied ¹⁴NO₂⁻, or other ¹⁴N intermediate (e.g. ¹⁴NO), from ammonia oxidation" and then explore the role for AnirK (reducing ¹⁵NO₂⁻) and AamoA (the ¹⁴N intermediate). However we will reference these 2 papers as they provide support for the nitrifier-denitrification like pathway we have invoked here and have edited the text to explicitly state NO. See lines 74, 233, 243-244, 254.

2. While gene abundances were used in this study to explain the nitrous oxide signals, utilization

of transcript would be more convincing, and perhaps yield more explanatory power over genes abundances alone. This is especially true for marine Thaumarchaea, which often display high activities at the base of the oxycline, but are often at abundance levels that are considerably lower. The metatranscriptomic analysis of the eastern tropical South Pacific oxygen minimum zone by Stewart et al. (2012) is an example that displays the mismatch between gene abundance and activity quite well.

Please see the initial response above

2. The choice of oligo-nucleotide primers used to assess the abundance of the amoA gene corresponding to both Bacterial and Thaumarchaeal groups is a bit surprising, given that both are utilized primarily in terrestrial environments, and the later is considerably degenerate. The primers developed by Mincer et. al. 2007 to target the Thaumarchaea are most commonly used. What was the specific justification for utilizing the primers used by Tourna et. al.?

We beg to differ with the reviewer here. Mincer has 262 citations, Tourna 272 citations in Web of Science, so the statement that Mincer are more widely used is clearly not correct. Both have clear provenance in the literature but Tourna provides good evidence of targeting amoA genes from both terrestrial and marine AOA. Therefore we have used a very well supported primer set that target the broadest group of AOA, not just known marine AOA.

4. (Minor) L128 "The overwhelming majority of studies have argued for nitrification by ammonia oxidizing bacteria ..." This is a weak and old argument by now.

The statement was redundant as it is covered at the end of the discussion section where we characterize our source of N₂O and to save space it has been removed from the discussion. This general line of argument was also raised by Reviewer 3 and we justify the structure of our introduction there in some detail, but in essence we seek to address the assumptions of a wide variety of potential readers, including the oceanographic community which has yet to assimilate the nuanced arguments within the microbial ecology community about the differences between AOA and AOB and their relative importance in N₂O production. Thus we present this entrenched, but recently challenged view that oceanic N₂O is produced by oxygen-stressed AOB. This is essential to properly place our work into a broad scientific context rather than assuming that a role for AOB has been conclusively disproven in the minds of all of our potential readers. Paragraph 2 and 3 within the Introduction do exactly this and we absolutely emphasise the now recognized importance of AOA over AOB in an OMZ. See paras 2 and 3 of the Introduction and lines 216 where this text has been removed but is now summed up on line 259-260.

References:

Kozlowski et al. (2016). Pathways and key intermediates required for obligate aerobic ammonia-dependent chemolithotrophy in bacteria and Thaumarchaeota ISME J. advanced online publication. Feb 16. doi: 10.1038/ismej.2016.2.

Martens-Habbena et al. (2015). The production of nitric oxide by marine ammonia-oxidizing archaea and inhibition of archaeal ammonia oxidation by a nitric oxide scavenger. Environ

Microbiol. 17: 2261-74.

Mincer et al. (2007) Quantitative distribution of presumptive archaeal and bacterial nitrifiers in Monterey Bay and the North Pacific Subtropical Gyre. *Environ Microbiol* 9: 1162-1175.

Stewart et al. (2012). Microbial metatranscriptomics in a permanent marine oxygen minimum zone. *Environ Microbiol.* 14: 23-40.

Reviewer #3 (Remarks to the Author):

The paper reports an extensive biogeochemical and microbiological study of rates and pathways of N₂O formation in the world's largest oxygen minimum zone (OMZ), concluding that N₂O production can be ascribed to ammonium-oxidizing Archaea through coupled nitrification-denitrification, and that rates can be predicted from a simple oxygen dependent rate expression. OMZs are recognized as important sources of atmospheric N₂O, but there is no consensus on the mechanisms and regulation of N₂O production there. Thus, if the conclusions hold, the study represents a substantial step forward, which should be of interest to a wide audience.

The strengths of the study include the high spatial coverage and large experimental dataset of good quality. As stated in the paper, it is the first detailed experimental investigation linking N₂O production under in situ conditions to a specific microbial pathway. In its present form, however, a number of issues appear to undermine the central conclusions.

1) The reported ¹⁵N-N₂O production rates greatly underestimate total N₂O production. N₂O production rates are based on incubations with ¹⁵N-labeled nitrite, a central finding being that the ¹⁵N-labeled N₂O formed is mainly of mass 45 (¹⁵N¹⁴N¹⁸O) rather than mass 46 (¹⁵N¹⁵N¹⁸O) as would be predicted from the isotopic composition of nitrite. This leads to the (correct) conclusion that N₂O is mainly stems from another source than nitrite and (l. 208) that the majority of N in N₂O is actually ¹⁴N - which implies that most of the N₂O produced during the experiments accumulated as ⁴⁴N₂O (¹⁴N¹⁴N¹⁸O). Based on the relative accumulation of masses 45 and 46 (Fig. 2c), typically (greater than or equal to) 50:1 but highly variable, and assuming random isotope pairing during the denitrification step to N₂O, we can estimate that total ⁴⁴N₂O production rates in the experiments were typically at least 10 times higher than ¹⁵N-N₂O and that this factor varied strongly between individual experiments (the situation is analogous to the use of the isotope pairing technique to quantify denitrification in aquatic sediments where a large fraction of N₂O/N₂ is formed through coupled nitrification-denitrification). This has serious implications: Firstly, the correlations of N₂O production to oxygen, gene numbers, etc., must apply to the total rates (or ⁴⁴N₂O rates) and not only to the ¹⁵N-N₂O rates as demonstrated now. It is not clear if this will be the case, because each data point should be scaled individually depending on the ⁴⁵N₂O/⁴⁶N₂O production ratio. Secondly, the rate expression used for modelling should also be based on total rates. As far as I can see, increasing the rates by an order of magnitude will lead to a similar increase in the modelled N₂O concentration (Fig. 5), which means that the model will no longer fit the measurements. The experimental dataset needs to be re-evaluated.

We think there's just been a simple mistake here. The reviewer begins by stating that "Based on the relative accumulation of masses 45 and 46 (Fig. 2c)", however Fig 2c did NOT show masses 45 and 46 i.e. $^{45}\text{N}_2\text{O}$ and $^{46}\text{N}_2\text{O}$, it showed the actual measured amount of $^{45}\text{N}_2\text{O}$ as a function of $^{45}\text{N}_2\text{O}$ predicted for simple denitrification with $^{15}\text{NO}_2^-$. (It is now redrawn to show $^{45}\text{N}_2\text{O}$ above that expected from denitrification but the message is the same). This tells us that the vast majority of N_2O cannot be due to denitrification because the measured frequency of $^{45}\text{N}_2\text{O}$ is far above that predicted for denitrification given the ^{15}N labelling of the NO_2^- pool (>86%). If N_2O were being produced solely through denitrification then, with random isotope pairing, the labelling of the N_2O would be binomially distributed relative to frequency of ^{14}N to ^{15}N in the NO_2^- pool and we would get $^{45}\text{N}_2\text{O}$ produced along the 1:1 line in Fig. 2c – which we do not. So it seems to us that the premise of the argument is not correct.

As a consequence, what we write on line 208 does not "impl(y)ies that most of the N_2O produced during the experiments accumulated as $^{44}\text{N}_2\text{O}$ " because the ratio of measured $^{45}\text{N}_2\text{O}$ to $^{46}\text{N}_2\text{O}$ cannot be explained by denitrification. We also think that the reviewer is wrong to "estimate that total $^{44}\text{N}_2\text{O}$ production rates in the experiments were typically at least 10 times higher than $^{15}\text{N}-\text{N}_2\text{O}$ ". If Fig 2c had shown both 45 and 46 you could not predict total N_2O from their ratio directly because the ratio of the two isotopes in the N_2O is not proportional to the labelling of the nitrite in the simple way implied.

We have, however, extended our calculations (equ. 3,4&5) in the methods (lines 365-390) to estimate total N_2O production from the total ^{14}N and $^{15}\text{N}-\text{NO}_2^-$ pool to make our data comparable to those working more frequently with ^{15}N and the production of N_2 but not as the reviewer suggested. We now use the complete form of the widely published calculations for estimating total rates of N_2 production by anammox and denitrification¹ and no longer just report the ^{15}N component. The overall production of N_2O that we can apportion to the reduction of exogenous NO_2^- is given by:

$$p\text{N}_2\text{O}_{\text{exogenous}} = p^{46}\text{N}_2\text{O} \times \text{FN}_{\text{NO}_2^-}^{-2} \quad (3)$$

where $\text{FN}_{\text{NO}_2^-}$ is the frequency of ^{15}N in the nitrite pool (Table S1). N_2O production that we cannot apportion to the reduction of exogenous NO_2^- i.e. through pairing with an alternative source of ^{14}N (e.g. NH_4^+ , NO , internal NO_2^- or other ^{14}N) which, for simplicity, we refer to as endogenous N_2O is given by:

$$p\text{N}_2\text{O}_{\text{endogenous}} = \text{FN}_{\text{NO}_2^-}^{-1} \times (p^{45}\text{N}_2\text{O} + 2 \times (1 - \text{FN}_{\text{NO}_2^-}^{-1}) \times p^{46}\text{N}_2\text{O}) \quad (4)$$

Finally, the total production of N_2O in our incubations with $^{15}\text{NO}_2^-$ and that which we use throughout for both the statistical analysis and parameterisation of the ID model reported here is the sum of the two former products:

$$p\text{N}_2\text{O}_{\text{total}} = p\text{N}_2\text{O}_{\text{endogenous}} + p\text{N}_2\text{O}_{\text{exogenous}} \quad (5)$$

Using this method we still show that total N_2O production ($p\text{N}_2\text{O}_{\text{total}}$ now used throughout) is still an exponential function of decreasing oxygen (Fig. 3 and S3) because it is highly correlated with the original $^{15}\text{N}-\text{N}_2\text{O}$ ($r^2 = 0.9845$, $P < 0.0001$). The use of total N_2O rather than just the $^{15}\text{N}-\text{N}_2\text{O}$ component has a negligible effect on the parameters derived from the non-linear mixed effects model (intercept, $a=120.6$ versus 132.4 and exponent, $b=0.051$ versus 0.048); especially b , the sensitivity to oxygen. The subsequent relationships with the archaeal genes (Fig. 4) are also unaffected and these new parameters have no effect on the output of the ID model (Fig. 5). Method lines 365-390.

Furthermore, it should be acknowledged that the experiments are, presumably, blind to "leaky" nitrifier N₂O production through the hydroxylamine pathway. Thus, total N₂O production could be even larger than what the calculations mentioned above will show. This could have been investigated by measurement of N₂O production in the ¹⁵N ammonium incubations, which were included in the study. Did the authors attempt such measurements?

The reviewer may have had a valid point here if we had any evidence of a significant contribution from AOB in these samples, which we do not. Furthermore, the Martens-Habben and Kozlowski papers referred to by Reviewer 2 shows a clear role for NO in AOA ammonia oxidation but one that is clearly distinct from any role NO may have in AOB. Therefore, it is very unlikely that leaky nitrifier N₂O is confounding our analysis in this study.

Please also note – that neither of the two concerns raised by the reviewer here would have any effect on either our revised or original story. We experimentally manipulated oxygen and directly measured an exponential increase in ¹⁵N-N₂O. The residual variation that couldn't be explained by oxygen correlates with archaeal gene abundance. Also, parameterizing a 1D model with our non-linear model coefficients reproduces the pattern of N₂O in the ocean (incredibly well!) and, with that model, we can balance direct estimates of air sea exchange – the issues raised by this reviewer do not change any of that.

2) The archaeal nirK gene numbers appear orders of magnitude too high. The central conclusion that N₂O is produced through archaeal nitrification-denitrification rests strongly on the correlation of rates and AnirK gene numbers determined by qPCR. These numbers are extremely high (Fig. S5), up to 10⁶ mL⁻¹, which is similar to or higher than total prokaryote counts in mesopelagic waters, and 2 orders of magnitude higher than the counts of archaeal amoA genes in the same samples. While the cells may have more than one copy of each gene, it is unlikely that they have 100 copies of nirK. Furthermore, a previous study (Lund et al. 2012) found a reasonable agreement between AnirK and AamoA in North Pacific waters, and typical numbers of 10⁴ mL⁻¹, in agreement with the AamoA counts here and in another study from the same OMZ (Beman et al. 2012, L&O). This discrepancy might suggest that the nirK primer set as applied here is not specific for thaumarchaeotal ammonium oxidizers, which clearly undermines the conclusion based on the correlation.

Please see the response to qPCR and molecular analysis above

3) The background of the paper, as presented here, is a bit of a strawman. The authors argue that the present understanding is that N₂O formation in OMZs is due to bacterial ammonium oxidizers. This is an out-dated point of view, which originates from the days before the role of Thaumarchaeota was realised. Studies by Francis, Beman, Stewart and others have clearly documented that bacteria play a very minor role in the ammonium oxidizer community and in ammonium oxidation in OMZs, and Santoro and coworkers have shown that Thaumarchaeota are a likely source of N₂O there, and that at least part of their N₂O production is through nitrification-denitrification. The novelty of the present story lies in the experimental approach and derivation of quantitative relationships, not in discovering the role of Thaumarchaeota.

As we state above in response to reviewer 2 there is still a substantial community of oceanographers and ocean modelers who have not assimilated the rapidly evolving ideas about AOA and AOB in ocean nitrogen cycling. As our data will be a challenge to the well-established

view of many researchers in this field we absolutely have to address the idea that AOB drive N₂O production in the oceans absolutely head on. However, we also focus a significant proportion of the original introduction on highlighting the growing recognition of the role that the archaea are playing in the production of N₂O including half of the references cited here. The intro starts with a broad and simple overview of N₂O in the ocean – and the multiple microbial metabolisms that may play a role. We then explore the thickening of OMZs and recognize that the perspective of ocean modelers is that N₂O is largely due to bacterial ammonia oxidation. The final section brings the story up to date by bringing in the archaea (which is the reviewer’s point above but it is not widely known to all with an interest in N₂O) and finish by simply stating that “There has, however, been no formal experimental characterisation of N₂O production at oxygen concentrations representative of the margins of an OMZ and/or the abundance of AOA (or any other candidate organism) in representative samples of the ocean². Here we provide experimental evidence....” We attempted to set the scene from all angles to broaden the appeal of our paper.

Specific comments

19: through 550 km is not informative specify inshore-offshore gradient or similar

As all of our sites were open-ocean with the transect running offshore we have changed this to “along a 550km offshore transect”. The full details of which are now clearly described in the legend and map in figure 1.

21: strong 45N₂O signature makes no sense without explanation of the methods

Agreed and have inserted ¹⁵NO₂⁻ higher up in the abstract but a full explanation is not possible with just 150 words. Line 19.

46: here and in many other places the authors use quotation marks in an unclear manner - does this indicate dubious results or what?

Removed as there use was a bit arbitrary on rereading the document.

48: explain "N₂O anomaly"

Defined, see line 47.

64: Thaumarchaeota is used elsewhere and should be correct

Corrected to Thaumarchaea throughout.

87: unclear sentence - how can the MLD extend to 25 m and then increase steadily?

Corrected and now reads “the mixed layer depth (MLD) extended down to approximately 20m to 25m and then density increased steadily to a sharp inflection at 35m to 40m, marking the base of the pycnocline (Fig. 1b).” Lines 88-90.

92: what is "the true anoxic core"??

This is a thorny issue but we have now gone with “the functionally anoxic core of the OMZ³” where the reference cited explains in detail that, although there may still be a handful of nano-molar amounts of oxygen (Thamdrup pers. comm), oxygen is, for all intents and purposes, functionally unavailable to aerobes and the core is therefore functionally anoxic. This has no

real bearing on our overall story and it is a passing line in the opening results describing the broader water column in the OMZ. Lines 94-96.

93: unclear: nitrite and N₂O minima were found at 350 m and occurred at 400-450 m??

The text has been edited to clarify this point “Deeper, at around 350m, oxygen became comparatively constant, with the functionally anoxic core of the OMZ³, where both the secondary nitrite maxima and N₂O minima were measured, occurring deeper still at 400m to 450m (Fig. S1a). lines 94-96.

Fig. S2 is very difficult to understand. Why are the same data points shown in multiple frames in both a and b? Why do the sections overlap sometimes but not always?

The co-plot function in R can be used to represent multivariate data but its usage might not be that familiar to everyone. Below are all the available datasets from transects giving the best coverage closest to ours in the tropical Pacific – we are MT in maroon, vertical -92.5W. Fig. S2 tries to show our profiles relative to all other profiles when the data are either sampled according to latitude or longitude. The programme also tries to sample or bin the data evenly – though the data are clearly not evenly distributed (e.g. between -100 to -130E). I have now reduced the overlap as much as possible but it is unavoidable in the longitude orientation because transect 130 and 8 slice through our transect from NW to SE and the bin, sampling either side of -100E, picks up our transect at 92.5W twice. Hope it is clearer now.

107: is there such a thing as O₂-free nitrogen gas? (and OFM is used but not defined anywhere)

No, in reality there probably isn't but oxygen-free-nitrogen (OFN) is just a common trade name of a routine gas. Changed to nitrogen on line 114 and the purity of our “OFN” is now defined in Table 1 where we provide a summary of our treatments.

108: dual labelled 45N₂O??

As above, reviewer 1.

120: exponential increase is confusing without stating the direction of oxygen concentrations - it

is an exponential decrease with increasing oxygen

We have modified the statement to “The overall exponential increase in production of N_2O with oxygen decreasing below $30 \mu\text{mol } O_2 L^{-1}$ is not only consistent with N_2O accumulating below $30 \mu\text{mol } O_2 L^{-1}$ in the water column (Fig. 1d) but also with distributions seen in many parts of the tropical North Pacific (as above, Fig. S2).” We have also edited this clause throughout the text to include decreasing. For example lines 18-19 “to decreasing oxygen between $1-30 \mu\text{mol } O_2 L^{-1}$ within and below the oxycline” and 129-130 and all others.

121: it is not obvious to me that the exponential function is consistent with the in situ distribution

The exponential response that we measure in a bottle is a pure biological response to decreasing oxygen, whereas the profile is an integral of both that biology and abiotic, physical factors. Both biology and physics shape the profile and that is what we capture, very well, with our 1D model (See new version in Fig. 5). The model is parameterized (M2, Table 2) to generate N_2O as a single biological response to decreasing oxygen which physics then distributes, shapes, to a steady-state profile in the water column. For example, diffusion through the pycnocline is slow, hence the accumulation of N_2O at the base of the pycnocline. As an OMZ is a consequence of rapid biology respiring oxygen and sluggish physics failing to replenish that oxygen – oxygen is slow in, N_2O (and CO_2) is slow out etc.

124: these developments are by no means "approximately linear". Two out of four clearly cease completely after 12 h, and the two others clearly decelerate after 20 h - the effect seems to be depth dependent. The underestimation of rates in the long incubations adds further uncertainty to issue #1 above.

See point 9 for reviewer 1. The last point is answered as part of the rebuttal to main point of criticism from reviewer 3 above i.e. we are not largely underestimating N_2O production.

200-2: please explain - why does the fact that nitrification is active lead the authors to expect no N_2O production from nitrite? Nitrifier denitrification depends on nitrification.

With all due respect the reviewer is citing this phrase out of context. The paragraph started by setting the scene on the previous line with “The overwhelming majority of studies have argued for nitrification by ammonia oxidizing bacteria (AOB) ($^{15}NH_4^+ \rightarrow ^{15}NH_2OH \rightarrow (^{15}NO + ^{15}N_2O) \rightarrow ^{15}NO_2^-$) as the principal source of oceanic N_2O ^{4,5}) but this view is being revised⁶. The potential for nitrification is clearly evident in our data and, that being so, then we would not expect any $^{15}N-N_2O$ production at all with $^{15}N-NO_2^-$ as the substrate.” As for reviewer 2, point 4, this section has been edited and the material is now covered elsewhere.

If the dominant source of N_2O was (classic) nitrification by ammonia oxidizing bacteria (as many who model oceanic N_2O production conceptualize it – hence the style of 30% of our Introduction and argument in the Discussion) then, with our added $^{15}NO_2^-$, we shouldn't have got any ^{15}N labeled N_2O because the ammonia oxidizing bacteria would be oxidizing NH_4^+ and not our NO_2^- . This is neither the case here nor in our previous work (Introduction) and, as we argue, there is no experimental evidence in the ocean for this pathway and we, like the others cited, couldn't even find the genes for any ammonia oxidizing bacteria. Yes, nitrifier denitrification depends on nitrification and that is indeed what we are proposing here. As we added $^{15}NO_2^-$ and measured predominantly $^{45}N_2O$ ($^{14}N + ^{15}N$), our ^{15}N was combined with a source of ^{14}N which we argue could be (but we, just like everyone else, don't know for sure) internally generated

¹⁴NO₂⁻ or other ¹⁴N intermediate of ammonia oxidation through some variant of the archaeal nitrification biochemical pathway - but not bacterial pathway. This is clearly covered in the revised text. Introduction lines 56-61 and main body of the Discussion 216-260.

209-11: the relevance of this sentence is not clear.

Yes, it did hang a bit and it has now been edited to link it better to the previous section. See lines 223-225.

225-6: the authors did not measure net nitrification (i.e. increase in concentrations of nitrite+nitrate), I believe. Furthermore, Kalvelage and coworkers have reported nitrate reduction to 20 μM oxygen in OMZ waters, so why would 1 μM exclude the process? The potential for nitrate being the source of ¹⁴N in N₂O should be discussed further.

An oversight by the reviewer perhaps as we did measure net nitrification; as was stated in the methods, results and the legend for Table S2 "...overall net nitrification from the accumulation of total ¹⁵NO_x⁻ (¹⁵NO₃⁻ combined with ¹⁵NO₂⁻)..." see lines 237-239. Hence we stand by our original assertion that as we did indeed measure net nitrification (nitrate production) there is no need for us to invoke nitrate reduction as a source of ¹⁴NO₂⁻ but have included the observations by Kalvelage et al for context (ref 34).

228-9: what does "and, though not directly" mean?

*This whole section has been rewritten in light of the corrected qPCR data and both *AamoA* and *AnirK* are now directly and equally represented in the multiple linear regression model. See new lines 240-245.*

240-1: this is pure speculation - being better adapted doesn't mean being perfect

We did not mean to imply perfect, and don't think that we do. This is one line of inference at the end of the proceeding argument. The point is to emphasize the need to recognize a major route of N₂O production as a real niche rather than as a consequence of simple oxygen stress (which drives N₂O production in AOB) and we would prefer to keep it in. It now appears on lines 254-257.

260: Babbin did not present a variant of the authors' model - it is the other way around

Appreciated and edited accordingly. See lines 275, 278 and 458

276-8: this seems to contradict the general applicability of the simple model advocated earlier

On reflection we do feel that this attempt to smooth out the differences between our model and Babbin's model is somewhat clumsy and really not required in this manuscript. Our simple model recovers the measured N₂O profile in the ETNP OMZ and Babbin's more complex model does not. So we have deleted the paragraph and left any discussion of the relative value of these models to elsewhere.

281-2: air-sea exchange rates should be independent of the assumptions about pathways of N₂O production?

We agree with the reviewer that air-sea exchange rates should be independent of the assumptions about pathways of N₂O production. Nevertheless, air-sea exchange depends on the

air-sea concentration gradient, so if the model over/under estimated N₂O production (regardless of the pathway) then the air-sea exchange may also be erroneous. Here our model agrees very well with our observations (Fig. 5). We have simplified this statement in the revised ms. in order to avoid confusion “The two methods that we used to estimate sea to air exchange (...) agreed very well with each other and with those in the literature - despite the different approaches.” See lines 285-287.

320: how were vials filled with He is they were only sealed later?

Thanks for spotting this, rewritten lines 245-248.

442: where does the Heaviside function come from? The data implies increasing N₂O production all the way to zero oxygen. What does a* refer to - it is not in the equation? (here again, why the quotation marks?)

The Heaviside function was used in the original Babbin et al. (2015) model as a switch for nitrification (nitrification is 'turned off' when O₂ < 0.4 μM). It is redundant in our model as the O₂ never decreased below this threshold in the upper 350 m of the water column. We now highlight this distinction between the original and our own variant of the Babbin model but the function has been removed from the revised ms. in order to avoid confusion. Lines 457-465.

Fig. 1: Specify definitions of the OMZ boundary and N deficit. It would be useful to have different markers for the different stations in c) and d).

We have added the respective definitions. We do, however, disagree with the reviewers' request for different markers on panels a, and b. The whole point of our approach is to not worry about the idiosyncrasies of the individual sites or profiles but rather to present an analysis of the entire dataset. Adding colour or symbols to a, and b, just adds noise and we feel that this would not be useful.

Fig. 2: Why are the highest values in panel b) not included in c)? E.g. two high outliers in treatments 3+4, one in treatment 5+6.

Pooling the data for treatments 3+4 and 5+6 changes the overall distribution of the data as we are, in effect, doubling the sample size from 12 to 24 in each case and that affects what is classed as an outlier in the box-plots. Since recalculating, to express our data as total N₂O production, in line with the reviewer, the overall distributions of the data in the boxplots have changed ever so slightly, the effect of treatment is now even stronger (e.g. Likelihood ratio test for treatment is now - d.f. 5, Chi-sq. 38.365, compared to 30.29) but our model output and conclusions remain the same.

Fig. 4: aomA should be amoA

Done, see new version of figure 4.

Table S2: The explanation of the rates is highly confusing. According to methods, nitrite oxidation was apparently from incubations with ¹⁵N nitrite. If nitrification is the sum of ammonium and nitrite oxidation, how can the values be lower than nitrite oxidation?

Net nitrification was measured as the net accumulation of ¹⁵NO_x⁻ during incubations with ¹⁵NH₄⁺ and nitrite oxidation was measured with ¹⁵NO₂⁻ so net nitrification is not the sum of ammonium and nitrite oxidation in the instance. Net nitrification relies on ammonium oxidation

whereas nitrite oxidation does not. Sorry if this wasn't clear and hope it isn't confusing any more. The important point is that none of this nitrification activity had the capacity to significantly influence the ratio of ^{15}N to ^{14}N in our incubations for the production of N_2O .

Table S5: Where are the results from the quantifications of non-candidate genes (bacterial 16S RNA etc.)? What is the justification for the choice of AnirK primers

Please see the response to qPCR and molecular analysis above.

- 1 Thamdrup, B. & Dalsgaard, T. Production of N_2 through anaerobic ammonium oxidation coupled to nitrate reduction in marine sediments. *Applied and environmental microbiology* **68**, 1312 - 1318, doi:10.1128/AEM.68.3.1312-1318.2002 (2002).
- 2 Löscher, C. R. *et al.* Production of oceanic nitrous oxide by ammonia-oxidizing archaea. *Biogeosciences* **9**, 2419-2429, doi:10.5194/bg-9-2419-2012 (2012).
- 3 Thamdrup, B., Dalsgaard, T. & Revsbech, N. P. Widespread functional anoxia in the oxygen minimum zone of the Eastern South Pacific. *Deep-Sea Research Part I- Oceanographic Research Papers* **65**, 36-45, doi:10.1016/j.dsr.2012.03.001 (2012).
- 4 Freing, A., Wallace, D. W. R. & Bange, H. W. Global oceanic production of nitrous oxide. *Philosophical Transactions of the Royal Society B-Biological Sciences* **367**, 1245-1255, doi:10.1098/rstb.2011.0360 (2012).
- 5 Nevison, C., Butler, J. H. & Elkins, J. W. Global distribution of N_2O and the Delta N_2O -AOU yield in the subsurface ocean. *Global Biogeochemical Cycles* **17**, doi:1119 10.1029/2003gb002068 (2003).
- 6 Santoro, A. E., Buchwald, C., McIlvin, M. R. & Casciotti, K. L. Isotopic Signature of N_2O Produced by Marine Ammonia-Oxidizing Archaea. *Science* **333**, 1282-1285, doi:10.1126/science.1208239 (2011).

Reviewers' comments:

Reviewer #1 (Remarks to the Author):

The authors have done a thorough and commendable job of addressing my concerns in the prior review, notably regarding the disconnect in the qPCR counts and clarifying the overall presentation of the experimental/sampling design and bottle effects. This is a solid experimental study on a potentially substantial, and somewhat overlooked, pathway to OMZ greenhouse gas production. There are many mechanistic details about this process that remain to be fleshed out. I suspect this paper will be a powerful motivator for such work.

Reviewer #3 (Remarks to the Author):

The authors have revised the paper extensively and have resolved many issues. My most serious concern remains, however, namely that the reported 15N-N₂O production rates greatly underestimate total N₂O production. This might in part be due to a couple of mistakes in my original comments for which I apologize. Although the error is not as large as I thought originally, I maintain that it might undermine the central conclusions.

Authors' comments:

"We think there's just been a simple mistake here. The reviewer begins by stating that "Based on the relative accumulation of masses 45 and 46 (Fig. 2c)", however Fig 2c did NOT show masses 45 and 46 i.e. 45N₂O and 46N₂O, it showed the actual measured amount of 45N₂O as a function of 45N₂O predicted for simple denitrification with 15NO₂⁻. (It is now redrawn to show 45N₂O above that expected from denitrification but the message is the same). This tells us that the vast majority of N₂O cannot be due to denitrification because the measured frequency of 45N₂O is far above that predicted for denitrification given the 15N labelling of the NO₂⁻ pool (>86%). If N₂O were being produced solely through denitrification then, with random isotope pairing, the labelling of the N₂O would be binomially distributed relative to frequency of 14N to 15N in the NO₂⁻ pool and we would get 45N₂O produced along the 1:1 line in Fig. 2c - which we do not. So it seems to us that the premise of the argument is not correct."

I made two mistakes in my original comment, which might have confused the authors, but the premise of my argument that "The reported 15N-N₂O production rates greatly underestimate total N₂O production" still holds, and the same applies to the newly calculated total N₂O production rates.

Regarding my mistakes:

(1) While Fig. 2c did not show 46N₂O production directly, it did so indirectly because the ordinate, 45N₂O predicted for denitrification, is directly related to 46N₂O production (Equation 1, l. 368). My estimate of the production ratio of 45N₂O to 46N₂O of 50:1 was wrong, however. If I understand l. 115 correctly (that 81% refers to the contribution of 45N₂O to 15N-N₂O; it can't refer to the relative increase above the expected value although that is what the text seems to imply - I didn't get this the 1st time), this ratio was on average 4:1 (0.81/(1-0.81)).

(2) Because the authors write in the discussion that "Put simply, the majority of N in the N₂O produced was actually 14N that was not derived from our 15N-NO₂⁻ tracer", I assumed that the authors agreed with me in the principles of isotope pairing that apply to N₂O formed through nitrifier-denitrification. I understand now that they don't and will therefore explain this issue in more detail:

Nitrifier-denitrification traditionally represents the reduction of nitrite via NO to N₂O (NH₄⁺ => NH₂OH => NO₂⁻ => NO => N₂O; e.g., Stein L.Y. 2011 in Nitrification, eds. Ward B.B. et al., ASM

Press) with the two steps being analogous to those found in canonical denitrifiers. The process is well-described in AOB. The authors argue that N₂O production in AOA occurs through a similar, if not identical pathway, which is agreement with the most recent literature. It is possible that NO is a free intermediate in ammonium oxidation in AOA, such that the mixing of N atoms originating from ammonium and nitrite, respectively occurs in the NO pool rather than (or as well as) in the nitrite pool (NH₄⁺ => NO; NO₂⁻ => NO; 2NO => N₂O). Regardless, the step leading to N₂O formation is NO reduction, as is the case in canonical and AOB nitrifier denitrification. This has important implications for the calculations of total N₂O production based on ¹⁵N-nitrite incubations as in the present case.

If N₂O forms from either nitrite or NO reduction, the nitrogen isotope composition of N₂O is the result of random isotope pairing during the reduction of 2NO to N₂O. This means that N₂O of masses 44, 45, and 46 will form at a ratio of (1-FNO)² : 2*FNO*(1-FNO) : FNO², where FNO represents the mol fraction of ¹⁵N in the NO pool. This further implies that even if FNO is not known, the total production of N₂O (masses 44 + 45 + 46) can be determined from the production of ⁴⁵N₂O and ⁴⁶N₂O as described by Nielsen (FEMS Ecol 86:357-62, 1992). This is the basis of Nielsen's well-established isotope pairing technique, which is widely used to determine denitrification rates in intact sediment cores. As I pointed out in my original comment, the situation in the sediment incubations is analogous to that in the present study: ¹⁵N-nitrate is added to the water column and reduced to N₂O (and ultimately N₂) in the sediment, and at some point during the process, the stream of "exogenous" ¹⁵N is diluted by ¹⁴N originating from nitrification. In fact, the 1st author has developed an elegant technique to determine both denitrification and anammox in sediment cores based on this principle.

Based on all this, I am very surprised that the authors in their new calculations of total N₂O production have chosen to treat nitrifier-denitrification as an anammox-type process (l. 375-383). Anammox is characterized by 1:1 (rather than random) pairing of N atoms originating from nitrite and ammonium, respectively. This pairing is brought about by the 1:1 reaction of NO and NH₄⁺ to form hydrazine (N₂H₄), which is unique to the highly specialized anammox bacteria. Anammox bacteria do not produce N₂O (it seems) and the authors provide no justification whatsoever for assuming the involvement of a similar, asymmetrical process in nitrifier denitrification. All available knowledge points to random isotope pairing.

Thus, the authors should either argue convincingly for a new pathway of N₂O formation with a unique 1:1 isotope pairing, which is what they have based their present calculations on, or recalculate their total N₂O production rates assuming the more realistic random isotope pairing expected for nitrifier denitrification. For a production ratio of ⁴⁵N₂O to ⁴⁶N₂O of 4:1 (as estimated above) the total rate will be about twice the rate of ⁴⁵N₂O production. This change is not as serious as I had anticipated in my original comments, but it is still substantial, and, as mentioned in my original comment, the ratio of ⁴⁵N₂O to ⁴⁶N₂O production seems to be highly variable, and recalculation will therefore not simply scale up the rates proportionally. Thus, I strongly disagree with the authors' statement:

"Please also note - that neither of the two concerns raised by the reviewer here would have any effect on either our revised or original story. We experimentally manipulated oxygen and directly measured an exponential increase in ¹⁵N-N₂O. The residual variation that couldn't be explained by oxygen correlates with archaeal gene abundance. Also, parameterizing a 1D model with our non-linear model coefficients reproduces the pattern of N₂O in the ocean (incredibly well!) and, with that model, we can balance direct estimates of air sea exchange - the issues raised by this reviewer do not change any of that."

The authors have addressed most of my other comments convincingly and I shall not spend more time on these until the rate issue has been resolved. Most importantly, the qPCR values have been corrected to realistic levels, although there still seems to be errors in the data presentation: In Fig. 4, Ln(abundance) values are {less than or equal to} 4, while the same data in Fig. S5 reaches up

to 9. Presumably Fig. 4 shows $\log(10)$ values. The averages of these numbers in Fig. S6 for AmoA and AnirK do not seem to match those given in the text (l. 155-6; e.g., $\ln(5.6e3)$ AmoA/mL = 8.6, yet the figure shows a value below 8). The problems with these numbers are not reassuring.

Reviewer #3 (Remarks to the Author):

The authors have revised the paper extensively and have resolved many issues. My most serious concern remains, however, namely that the reported ^{15}N - N_2O production rates greatly underestimate total N_2O production. This might in part be due to a couple of mistakes in my original comments for which I apologize. Although the error is not as large as I thought originally, I maintain that it might undermine the central conclusions.

Authors' comments:

"We think there's just been a simple mistake here. The reviewer begins by stating that "Based on the relative accumulation of masses 45 and 46 (Fig. 2c)", however Fig 2c did NOT show masses 45 and 46 i.e. $^{45}\text{N}_2\text{O}$ and $^{46}\text{N}_2\text{O}$, it showed the actual measured amount of $^{45}\text{N}_2\text{O}$ as a function of $^{45}\text{N}_2\text{O}$ predicted for simple denitrification with $^{15}\text{NO}_2^-$. (It is now redrawn to show $^{45}\text{N}_2\text{O}$ above that expected from denitrification but the message is the same). This tells us that the vast majority of N_2O cannot be due to denitrification because the measured frequency of $^{45}\text{N}_2\text{O}$ is far above that predicted for denitrification given the ^{15}N labelling of the NO_2^- pool (>86%). If N_2O were being produced solely through denitrification then, with random isotope pairing, the labelling of the N_2O would be binomially distributed relative to frequency of ^{14}N to ^{15}N in the NO_2^- pool and we would get $^{45}\text{N}_2\text{O}$ produced along the 1:1 line in Fig. 2c - which we do not. So it seems to us that the premise of the argument is not correct."

I made two mistakes in my original comment, which might have confused the authors, but the premise of my argument that "The reported ^{15}N - N_2O production rates greatly underestimate total N_2O production" still holds, and the same applies to the newly calculated total N_2O production rates.

Regarding my mistakes:

(1) While Fig. 2c did not show $^{46}\text{N}_2\text{O}$ production directly, it did so indirectly because the ordinate, $^{45}\text{N}_2\text{O}$ predicted for denitrification, is directly related to $^{46}\text{N}_2\text{O}$ production (Equation 1, l. 368). My estimate of the production ratio of $^{45}\text{N}_2\text{O}$ to $^{46}\text{N}_2\text{O}$ of 50:1 was wrong, however. If I understand l. 115 correctly (that 81% refers to the contribution of $^{45}\text{N}_2\text{O}$ to ^{15}N - N_2O ; it can't refer to the relative increase above the expected value although that is what the text seems to imply - I didn't get this the 1st time), this ratio was on average 4:1 ($0.81/(1-0.81)$).

(2) Because the authors write in the discussion that "Put simply, the majority of N in the N_2O produced was actually ^{14}N that was not derived from our ^{15}N - NO_2^- tracer", I assumed that the authors agreed with me in the principles of isotope pairing that apply to N_2O formed through nitrifier-denitrification. I understand now that they don't and will therefore explain this issue in more detail:

Nitrifier-denitrification traditionally represents the reduction of nitrite via NO to N_2O ($\text{NH}_4^+ \Rightarrow \text{NH}_2\text{OH} \Rightarrow \text{NO}_2^- \Rightarrow \text{NO} \Rightarrow \text{N}_2\text{O}$; e.g., Stein L.Y. 2011 in Nitrification, eds. Ward B.B. et al., ASM Press) with the two steps being analogous to those found in canonical denitrifiers. The process is well-described in AOB. The authors argue that N_2O production in AOA occurs through a similar, if not

identical pathway, which is agreement with the most recent literature. It is possible that NO is a free intermediate in ammonium oxidation in AOA, such that the mixing of N atoms originating from ammonium and nitrite, respectively occurs in the NO pool rather than (or as well as) in the nitrite pool ($\text{NH}_4^+ \Rightarrow \text{NO}$; $\text{NO}_2^- \Rightarrow \text{NO}$; $2\text{NO} \Rightarrow \text{N}_2\text{O}$). Regardless, the step leading to N_2O formation is NO reduction, as is the case in canonical and AOB nitrifier denitrification. This has important implications for the calculations of total N_2O production based on ^{15}N -nitrite incubations as in the present case.

If N_2O forms from either nitrite or NO reduction, the nitrogen isotope composition of N_2O is the result of random isotope pairing during the reduction of 2NO to N_2O . This means that N_2O of masses 44, 45, and 46 will form at a ratio of $(1-\text{FNO})^2 : 2*\text{FNO}*(1-\text{FNO}) : \text{FNO}^2$, where FNO represents the mol fraction of ^{15}N in the NO pool. This further implies that even if FNO is not known, the total production of N_2O (masses 44 + 45 + 46) can be determined from the production of $^{45}\text{N}_2\text{O}$ and $^{46}\text{N}_2\text{O}$ as described by Nielsen (FEMS Ecol 86:357-62, 1992). This is the basis of Nielsen's well-established isotope pairing technique, which is widely used to determine denitrification rates in intact sediment cores. As I pointed out in my original comment, the situation in the sediment incubations is analogous to that in the present study: ^{15}N -nitrate is added to the water column and reduced to N_2O (and ultimately N_2) in the sediment, and at some point during the process, the stream of "exogenous" ^{15}N is diluted by ^{14}N originating from nitrification. In fact, the 1st author has developed an elegant technique to determine both denitrification and anammox in sediment cores based on this principle.

Based on all this, I am very surprised that the authors in their new calculations of total N_2O production have chosen to treat nitrifier-denitrification as an anammox-type process (l. 375-383). Anammox is characterized by 1:1 (rather than random) pairing of N atoms originating from nitrite and ammonium, respectively. This pairing is brought about by the 1:1 reaction of NO and NH_4^+ to form hydrazine (N_2H_4), which is unique to the highly specialized anammox bacteria. Anammox bacteria do not produce N_2O (it seems) and the authors provide no justification whatsoever for assuming the involvement of a similar, asymmetrical process in nitrifier denitrification. All available knowledge points to random isotope pairing.

1. Thus, the authors should either argue convincingly for a new pathway of N_2O formation with a unique 1:1 isotope pairing, which is what they have based their present calculations on, or recalculate their total N_2O production rates assuming the more realistic random isotope pairing expected for nitrifier denitrification. For a production ratio of $^{45}\text{N}_2\text{O}$ to $^{46}\text{N}_2\text{O}$ of 4:1 (as estimated above) the total rate will be about twice the rate of $^{45}\text{N}_2\text{O}$ production. This change is not as serious as I had anticipated in my original comments, but it is still substantial, and, as mentioned in my original comment, the ratio of $^{45}\text{N}_2\text{O}$ to $^{46}\text{N}_2\text{O}$ production seems to be highly variable, and recalculation will therefore not simply scale up the rates proportionally. Thus, I strongly disagree with the authors' statement:

"Please also note - that neither of the two concerns raised by the reviewer here would have any effect on either our revised or original story. We experimentally manipulated oxygen and directly measured an exponential increase in ^{15}N - N_2O . The residual variation that couldn't be explained by oxygen correlates with archaeal gene abundance. Also, parameterizing a 1D model with our non-linear model coefficients reproduces the pattern of N_2O in the ocean (incredibly well!) and, with that model, we can balance direct estimates of air sea exchange - the issues raised by this reviewer do not change any of that."

We must admit that we had not fully considered this point of view and its inclusion in the manuscript will ultimately strengthen its impact and broaden its appeal. However, we believe, and argue below, and in the text, that both routes are possible; yet our data, modelling and literature support our favored 1:1 route.

We begin with the literature in support of our 1:1 mechanism of N_2O production. The most recent paper by Kozłowski et al 2016¹ provides two models: one with a 1:1 pairing between NH_2OH and NO for the *Thaumarchaeota* (a, below); and one for the more classic $NO + NO$ known for bacterial nitrifier-denitrification (b, below). Hence the reviewer's view that "All available knowledge points to random isotope pairing" needs revising but there is far more substance to our argument that this. This point of view is included at lines 69-75, 233-236, 245-248, 256-259-262. Methods 381, 389-394.

We have now added calculations 6 & 7 to the Methods (lines 389-394) to represent this more classic pathway as suggested by the reviewer. This gives us TWO variants of total N_2O production: where equations 1 to 5 generate our favored pN_2O_{total} and where around 81% comes from a 1:1 coupling (as in a above) and equations 6 & 7 generate the $NO+NO$ alternative pN_2O_{total}' (b above). The two are strongly related to each other (below where $P < 0.0001$, $r^2 = 0.7339$, $d.f. = 68$). Hence, despite the reviewer's doubts that they would not simply scale with each other – they do.

We have now run both variants through our non-linear-mixed-effects model. Our original model is **M2**, as it was before, and the new variant for pN_2O_{total}' is now **M5** in Table 2. You can see quite clearly from the figure above and Table 2 that **M5** generates a higher intercept 'a'; though the shape of the response to oxygen 'b' is not that different: **M2** $a = 120$, $b = 0.0514$; **M5** $a = 613$, $b = 0.0681$.

We then use both pairs of coefficients to parameterize our 1D-model and use that to reproduce the profile of N_2O in the ocean (Fig. 5b). The new figure 5 is also reproduced below for ease:

The difference between either calculation is, admittedly, quite subtle but **M5** (i.e. $NO+NO pN_2O_{total}'$) over estimates production of N_2O throughout the oxycline, whereas our favored route, parameterized using **M2**, matches the profile incredibly well. Quite simply, therefore, we propose that our 1:1

calculation, parameterized in **M2**, matches the measured profile better than the alternative **M5** (i.e. $\text{NO}+\text{NO pN}_2\text{O}_{\text{total}}$) and is therefore our preferred mode of production. See lines 181-190, full discussion 233-242, Methods 462, the revised legend for Fig. 5 and title for Table 2.

Additional support for our proposed route surely comes from the fact that a classic mode of bacterial nitrifier-denitrification i.e. $\text{NO}+\text{NO}$ (**M5**), would require some bacterial, nitrifier-denitrifier genomic potential but, as we state in the text, “Here we were not able to detect the bacterial ammonia mono-oxygenase genes, β -amoA or γ -amoA. Given this apparent absence of any bacterial, nitrifier-denitrifier genomic potential, along with the overestimation of N_2O production in our model (**M5**, Fig. 5b), through such a path, we would refute bacterial nitrifier-denitrification in this setting.” See Discussion lines 233-242.

Not only could we find no bacterial, nitrifier-denitrifier genomic potential to support the classic path but our statistical modelling quite clearly shows that inclusion of our two archaeal gene abundances explains our measured production of N_2O better than oxygen alone (Fig 4 and Table S3). If we are going to include a robust statistical analysis of our data then it would quite simply be wrong to ignore the influence of archaeal gene abundance on the measured non-linear production of N_2O . Given that a 1:1 coupling in the archaea is now accepted in the literature and that such a route provides the most parsimonious explanation of our data, we therefore believe that what we present is the best explanation of our data. We do, however, really appreciate the effort the reviewer has expended here, as the inclusion of a classic bacterial mode of nitrifier – denitrification in Fig 5b (**M5**), that we can show overestimates production clearly demonstrates that we are looking at something novel here and for that we must thank them especially.

2. The authors have addressed most of my other comments convincingly and I shall not spend more time on these until the rate issue has been resolved. Most importantly, the qPCR values have been corrected to realistic levels, although there still seems to be errors in the data presentation: **a.** In Fig. 4, $\text{Ln}(\text{abundance})$ values are {less than or equal to} 4, while the same data in Fig. S5 reaches up to 9. Presumably Fig. 4 shows $\log(10)$ values.

The figure legend and axes' titles clearly state that the data have been centered (e.g. Ln AnirK_c ; as is correct for multiple linear regression) and how this was done is described in the methods “then more rigorously using multiple regression and the entire, linearized and centered (x_c) data-set (natural log, $x_c = x - x_{\text{mean}}$, $n=70$).” So there is nothing wrong with the values - See lines 508-511 and the legend for Fig. 4. There was an axis typo in figure S5 where ‘log’ should have read ‘ln’ which has now been corrected and may have led to this confusion.

b. The averages of these numbers in Fig. S6 for AmoA and AnirK do not seem to match those given in the text (l. 155-6; e.g., $\text{Ln}(5.6\text{e}3)$ AmoA/mL = 8.6, yet the figure shows a value below 8). The problems with these numbers are not reassuring.

There is nothing wrong with the values per se it is just the way the mean was calculated: the means of the original linear data are given in the text referred to by the reviewer, yet the mean of the ln values was generated automatically when graphing the data – this has been corrected by manually entering and graphing the ln value of each mean given in the text but it has no effect on anything we say. See new Fig. S6.

- 1 Kozlowski, J. A., Stieglmeier, M., Schleper, C., Klotz, M. G. & Stein, L. Y. Pathways and key intermediates required for obligate aerobic ammonia-dependent chemolithotrophy in bacteria and Thaumarchaeota. *ISME J*, doi:10.1038/ismej.2016.2 (2016).

REVIEWERS' COMMENTS:

Reviewer #3 (Remarks to the Author):

The revision resolves the central issue that I raised in my previous evaluation concerning how total N₂O production is calculated from the production of ¹⁵N-labeled N₂O. Based on a recent publication (Kozłowski et al. 2016), the authors assume that N₂O production by the dominating ammonium oxidizers in the system, the Thaumarchaeota (I believe this is the accepted name and not "Thaumarchaea" as used in the paper – e.g. Brochet-Armanet Nature Rev Microbiol 6:245) is through a hybrid pathway that combines N from two different sources (presumably NH₂OH and NO). In this case, the rate calculation used in the previous version is compelling (see potential minor issues below), and the conclusions are therefore also robust. Furthermore, an alternative approach to calculating the rates is now also presented, and it also provides a reasonable fit to the data, which demonstrates that the conclusions are not strongly dependent on the pathway assumption. Some other minor issues have also been clarified.

Only a few issues require attention now:

Mainly, the methods section describing the calculation of N₂O rates should be restructured. It looks like a new paragraph was simply added to the old text, but it would make more sense to start the section (l. 373-403) by stating that N₂O was calculated in two different ways: 1) assuming hybrid N₂O formation and attributing ⁴⁶N₂O production to denitrification (? – this also needs to be clarified, and I am not sure that mentioning the Arabian Sea results is a big help to the reader) and 2) assuming that all N₂O forms from a denitrification-type pathway with random isotope pairing. The reasoning for model 1 (now in l. 398-9) should be mentioned from the beginning. The authors also need to comment on how ⁴⁶N₂O may form (i.e. the "exogenous" N₂O in eqn. 5). According to Kozłowski, Thaumarchaeota do not have nitric oxide reductase, Nor, and therefore cannot produce N₂O from nitrite alone. Is ⁴⁶N₂O from canonical denitrification, then? In any case, it seems that the N₂O production used to correlate to Thaumarchaeotal gene abundances in Fig. 4 b and c should only be N₂O(endogenous). These issues can be discussed in the methods or in the supplement.

l. 242-244: I can't follow this argument. The fact that there is a net nitrification doesn't exclude the co-occurrence of nitrate reduction and nitrite oxidation, which, if gross rates are high enough, could lead to an equilibration in the labelling of nitrite and nitrate, and thereby a substantial dilution of ¹⁵N-nitrite.

l263-4: I am lost here. According to Kosłowski, hybrid N₂O formation could be a spontaneous process, and there is no indication of a link to energy conservation. The sentence refers to ref. 36 on AOB Nitrosomonas, which uses a different pathway, and there is no evidence of energy conservation from N₂O formation in fungi, as far as I know.

Fig S5a: error in x-axis title.

A general note: The paper now refers to the Thaumarchaeotal N₂O pathway as similar to co-denitrification. I would recommend using the term hybrid N₂O formation (at least in the abstract and introduction), which is used by Kozłowski and seems to be catching on rapidly at conferences.

Reviewer #3 (Remarks to the Author):

The revision resolves the central issue that I raised in my previous evaluation concerning how total N₂O production is calculated from the production of ¹⁵N-labeled N₂O. Based on a recent publication (Kozlowski et al. 2016), the authors assume that N₂O production by the dominating ammonium oxidizers in the system, the Thaumarchaeota (I believe this is the accepted name and not "Thaumarchaea" as used in the paper – e.g. Brochet-Armanet Nature Rev Microbiol 6:245) is through a hybrid pathway that combines N from two different sources (presumably NH₂OH and NO).

We have adopted Thaumarchaeota throughout the revised manuscript e.g. lines 68, 73, 272, 274, 278 and 404.

In this case, the rate calculation used in the previous version is compelling (see potential minor issues below), and the conclusions are therefore also robust. Furthermore, an alternative approach to calculating the rates is now also presented, and it also provides a reasonable fit to the data, which demonstrates that the conclusions are not strongly dependent on the pathway assumption. Some other minor issues have also been clarified.

Only a few issues require attention now:

Mainly, the methods section describing the calculation of N₂O rates should be restructured. It looks like a new paragraph was simply added to the old text, but it would make more sense to start the section (l. 373-403) by stating that N₂O was calculated in two different ways: 1) assuming hybrid N₂O formation and attributing ⁴⁶N₂O production to denitrification (? – this also needs to be clarified, and I am not sure that mentioning the Arabian Sea results is a big help to the reader) and 2) assuming that all N₂O forms from a denitrification-type pathway with random isotope pairing. The reasoning for model 1 (now in l. 398-9) should be mentioned from the beginning.

We have completely rewritten this section of the methods to introduce each respective method for calculating Total N₂O production at the start of this section. See lines 398-462.

The authors also need to comment on how ⁴⁶N₂O may form (i.e. the "exogenous" N₂O in eqn. 5). According to Kozlowski, Thaumarchaeota do not have nitric oxide reductase, Nor, and therefore cannot produce N₂O from nitrite alone. Is ⁴⁶N₂O from canonical denitrification, then?

The reviewer is absolutely right. On average, 19% of our N₂O production can be accounted for by simple reduction of NO₂⁻, through NO to N₂O but, because the archaea apparently lack the Nor to do that, we, and the reviewer, have to assume this to be canonical, bacterial denitrification, at least to N₂O as we measured no production of N₂. We thought this implicit in

our argument for equations 1 to 5 but now state it more explicitly on lines 420, 409 and 434 and see discussion lines 282-288. However, just because the molecular community has not found a Nor coding gene in laboratory strains of archaea, does not preclude an alternative being present in the ocean. In effect, all of the N₂O could actually be produced by the archaea but this full argument is probably better suited to an entirely separate paper.

In any case, it seems that the N₂O production used to correlate to Thaumarchaeotal gene abundances in Fig. 4 b and c should only be N₂O(endogenous). These issues can be discussed in the methods or in the supplement.

We appreciate this point of view but it is not as simple as this. Figure 4 is a visual presentation of the multiple linear regression used to explore any potential relationships between the production of N₂O, oxygen, and all four of the functional genes for the bacteria and archaea. Panels 4b and 4c are, in effect, the residuals in N₂O production, after oxygen has been accounted for, as a function of the two archaeal gene abundances i.e. the output from model 14, Supplementary Table 3. It is the very fact that we see this relationship with Total N₂O production and the archaea genes that forms the basis of our argument i.e. the 1:1 archaea hybrid formation dominates the production of N₂O. If we separate out the analysis for endogenous N₂O then we would need to do the same for the exogenous N₂O as that should only be produced by the bacteria but we think this would be an unnecessary over-complication of our main story.

1. 242-244: I can't follow this argument. The fact that there is a net nitrification doesn't exclude the co-occurrence of nitrate reduction and nitrite oxidation, which, if gross rates are high enough, could lead to an equilibration in the labelling of nitrite and nitrate, and thereby a substantial dilution of ¹⁵N-nitrite.

The reviewer is right in that our presentation of the argument against nitrate reduction was an oversimplification. Having now been through and fully analysed all of the Kalveladze¹ data we can still argue that nitrate reduction would have had no significant effect on our interpretation of our data. For example, the entire Kalveladze dataset (below – Top figure) does indeed show simultaneous oxidation of nitrite and reduction of nitrate up to ~16 μM O₂ but the very high rates of nitrate reduction tend to occur when oxygen is below our lowest concentration of ~ 1 μM O₂. Although we didn't attempt to measure nitrate reduction, our average rate of nitrite oxidation of 19 nmol N L⁻¹ d⁻¹ agrees very well with theirs of 14 nmol N L⁻¹ d⁻¹ which, most importantly, is practically the same as their median rate for nitrate reduction of 21 nmol N L⁻¹ d⁻¹. If we zoom in on the data (below – bottom figure), we can see that activity of this magnitude for both processes would indeed be possible at oxygen concentrations where we measured most production of N₂O i.e., between 1 and <30 μM O₂. Nitrate reduction at 21 nmol N L⁻¹ d⁻¹ would still only dilute our ¹⁵NO₂⁻ pool (10000 nM) by some 0.6%; whereas it would need to have been diluted by ~200% in order to generate our average ratio of ⁴⁵N₂O to ⁴⁶N₂O. We have edited the text to include this additional argument, see new lines 256-263.

1263-4: I am lost here. According to Koslowski, hybrid N₂O formation could be a spontaneous process, and there is no indication of a link to energy conservation. The sentence refers to ref. 36 on AOB Nitrosomonas, which uses a different pathway, and there is no evidence of energy conservation from N₂O formation in fungi, as far as I know.

Having now read this again we can see where the confusion lay and have edited the text accordingly and removed any reference to fungi. The current point of view is that the archaea may use some hybrid mechanism to produce N₂O which, if spontaneous, wouldn't itself be linked to energy conservation but we can argue that the production of the likely precursor substrates NH₂OH and NO would. We have edited the text to clarify and tone down our claim on lines 282-285.

The overall argument may, however, not be that straightforward. For example, we simply measured an exponential increase in N₂O production as a function of declining oxygen and then,

in effect, correlated the residuals in that relationship to the abundance of two archaeal functional genes. The fact that ~81%, on average, of that N₂O was single labelled ⁴⁵N₂O leads us down the path of a 1:1 coupling and, given the accepted point of view of Koslowski, to the spontaneous hybrid formation of N₂O. That spontaneous production would need to be perfectly aligned with the production of the precursors and non-saturating to generate our exponential patterns. Further, no organisms is likely to want either NH₂OH or NO accumulating near it, so it is at least plausible that a biological mode of reducing the two to N₂O has evolved in the ocean that may not be present in the Koslowski laboratory strains.

Fig S5a: error in x-axis title.

Thanks for spotting this: oxygen is now oxygen.

A general note: The paper now refers to the Thaumarchaeotal N₂O pathway as similar to co-denitrification. I would recommend using the term hybrid N₂O formation (at least in the abstract and introduction), which is used by Kozlowski and seems to be catching on rapidly at conferences.

We have adopted hybrid N₂O formation throughout and included further discussion of the potential routes of N₂O production in the Methods in line with this and the comments above. See new lines 26, 79, 283, 292, 333, 337, 404 and 424.

- 1 Kalvelage, T. *et al.* Nitrogen cycling driven by organic matter export in the South Pacific oxygen minimum zone. *Nature Geoscience* **6**, 228-234, doi:10.1038/ngeo1739 (2013).